# Municipal Solid Waste Thermal Analysis—Pyrolysis Kinetics and Decomposition Reactions

**Ewa Syguła** , **Kacper Świechowski** , **Małgorzata Hejna, Ines Kunaszyk and Andrzej Białowiec ***

Department of Applied Bioeconomy, Wrocław University of Environmental and Life Sciences,
37a Chełmońskiego Str., 51-630 Wrocław, Poland; ewa.sygula@upwr.edu.pl (E.S.);
kacper.swiechowski@upwr.edu.pl (K.Ś.); 113485@student.upwr.edu.pl (M.H.); ines.kunaszyk@gmail.com (I.K.)
* Correspondence: andrzej.bialowiec@upwr.edu.pl

**Abstract:** In this study, 12 organic waste materials were subjected to TG/DTG thermogravimetric analysis and DSC calorimetric analysis. These analyses provided basic information about thermochemical transformations and degradation rates during organic waste pyrolysis. Organic waste materials were divided into six basic groups as follows: paper, cardboard, textiles, plastics, hygiene waste, and biodegradable waste. For each group, two waste materials were selected to be studied. Research materials were (i) paper (receipts, cotton wool); (ii) cardboard (cardboard, egg carton); (iii) textiles (cotton, leather); (iv) plastics (polyethylene (PET), polyurethane (PU)); (v) hygiene waste (diapers, leno); and (vi) biodegradable waste (chicken meat, potato peel). Waste materials were chosen to represent the most abundant waste that can be found in the municipal solid waste stream. Based on TG results, kinetic parameters according to the Coats–Redfern method were determined. The pyrolysis activation energy was the highest for cotton, 134.5 kJ $\times$ (mol·K)$^{-1}$, and the lowest for leather, 25.2 kJ $\times$ (mol·K)$^{-1}$. The DSC analysis showed that a number of transformations occurred during pyrolysis for each material. For each transformation, the normalized energy required for transformation, or released during transformation, was determined, and then summarized to present the energy balance. The study found that the energy balance was negative for only three waste materials—PET ($-220.1$ J $\times$ g$^{-1}$), leather ($-66.8$ J $\times$ g$^{-1}$), and chicken meat ($-130.3$ J $\times$ g$^{-1}$)—whereas the highest positive balance value was found for potato peelings (367.8 J $\times$ g$^{-1}$). The obtained results may be applied for the modelling of energy and mass balance of municipal solid waste pyrolysis.

**Keywords:** TGA; DTG; DSC; thermogravimetric analysis; differential scanning calorimetry; municipal solid waste; organic waste; proximate analysis; process kinetics; Coats–Redfern method

## 1. Introduction

### 1.1. Background

The management of municipal solid waste (MSW) relates to the efficiency and optimization of the process, in addition to the sustainable use of products. Effective waste management methods focus not only on minimizing the volume of waste or eliminating sanitary hazards. An important element in the choice of waste treatment technology is the cost effectiveness of the process [1]. In the case of thermal waste conversion, the volume of waste is significantly reduced or incinerated completely, and high process temperatures ensure the hygienization of waste [2]. Thermal processes, including low-temperature pyrolysis, are associated with the supply of energy to the process and its recovery in the form of heat or products such as gas, oil, or biochar. The selection of substrates for the process of low-temperature pyrolysis relates to thermal characteristics of waste, combustion heat, or basic technical analysis determining the content of ash, volatile matter, and fixed carbon [3]. Compared to other thermal methods, such as combustion or gasification, pyrolysis requires energy input but is a significantly more environmentally safe process. MSW may contain hazardous organic substances, pathogens, or parasites. It is also a source of odor

emission and has a rotting tendency. As a result, its storage and handling are challenging. These issues can be overcome using low-temperature pyrolysis, which converts MSW into biochar or carbonized solid fuel (CSF), depending on use purposes [4]. Furthermore, the study of co-pyrolysis of biomass with MSW in a $CO_2$ environment has shown a positive environmental effect. Co-pyrolysis results in an increased quantity of flammable gases, and a decrease in tar production and harmful volatile organic compounds emission [5,6].

Organic waste accounts for about 77% of the MSW stream generated globally, comprising food waste and green waste 44%, plastics 12%, and paper 17%. EU legislation aims to reduce the MSW stream going to landfill to below 10% by 2035 [7]. Alternatively, waste going to landfills can be managed in another waste management sector, including thermal waste conversion processes to fuels that are easy to transport and store. The implementation of local pyrolysis plants contributes significantly to the reduction of greenhouse gas emissions [8]. Low-temperature pyrolysis, which is carried out at temperatures up to 500 °C, mainly initiates the formation of solid products in the form of biochar. The product of low-temperature pyrolysis can be used not only as a fuel, but also as a soil additive for the improvement of crop productivity and carbon neutrality [9,10].

### 1.2. Importance of Thermal Analysis

MSW pyrolysis is a highly complex process, and its results are difficult to predict. Municipal solid waste is a mixture of different materials with different chemical and physical features. Moreover, MSW composition depends on the production location (country, large city, small village), and season. Because each material is different, they have varied thermal degradation characteristics. As a result, for each MSW component, different process conditions (process temperature, heating rate, process duration, particle size, etc.) are needed to obtain the desired product. Depending on the process objective, the gas, liquid, or solid fraction is maximized using appropriate process conditions and technologies. For the description of pyrolysis, kinetics parameters are used. The kinetic parameters are used to compare different material and their thermal behavior for the prediction of process results. These parameters are determined using experimental data obtained at a laboratory scale.

### 1.3. Thermal Analysis Methods

According to the International Confederation for Thermal Analysis and Calorimetry (ICTAC), thermal analysis is a set of research methods that provide information about the relationship between a test sample and its temperature under controlled heating or cooling conditions [11,12]. Thermal methods are used to study chemical reactions and phase transformations that take place in the material due to temperature changes. It is possible to determine the kinetic or thermodynamic parameters of the process. The thermal parameters (glass transition, melting, decomposition, poly-morphic transformation, heat of fusion, crystallization, polymorphic transformation, and specific heat) provide information about the properties of the materials studied and about how they were produced [13].

TGA thermal analysis is used to determine changes in physical and chemical properties at an increasing temperature as a function of time, taking into account the mass loss of the sample [14]. The physical transformations observed during a TGA study are second-order phase transformations, desorption, or evaporation. In the case of chemical transformations, changes related to oxidation degradation, decomposition, and loss of volatile organic compounds (VOCs) are observed [15].

Differential scanning calorimetry (DSC) is based on the principle of measuring heat intensity or, more precisely, the difference in flux between the test and reference samples during thermal transformations [14]. The DSC method is used, among other things, for characterization of materials, stability studies, phase diagram evaluation, kinetics studies, or determination of heat capacity. DSC is dependent on measurement parameters such as heating rate, temperature, and sample size [16]. The measurement curve defines peaks that are assigned to exothermic and endothermic reactions. The above information enables the study of the energy balance of materials during waste thermal treatment [17].

The combined methods of thermogravimetric analysis (TGA) and differential scanning calorimetry (DSC) mainly report the changes in heat intensity due to ongoing chemical reactions, in combination with mass changes. Parameters related to phase transformations, thermal stability, and latent heat enthalpy are critical in standardizing material properties in the long term [18].

A large number of methods have been developed for the determination of kinetics parameters. One of these used in this study is the Coats–Redfern method. This method is based on the Arrhenius equation. This method is used to assess the validity of the model and ensures that the best model of the distribution response is fitted. Analysis of kinetics using the Coats–Redfern method determines the activation energies and the pre-exponential factor, which are the dominant factors in the reactivity equations. The pre-exponential factor is closely related to the structure of the material, whereas the activation energy characterizes the sensitivity of the material to temperature, which translates into the rate of the reactions occurring [19,20].

### 1.4. Research Aim

This study aimed to analyze the thermal characteristics of the selected components of six groups of MSW: paper (fiscal receipts and cotton wool); cardboard (grey cardboard and egg cartons); textiles (cotton and natural leather); plastics (polyethylene terephthalate (PET) and polyurethane (PU)); hygienic waste (pampers and leno); and biodegradable waste (meat and potato peelings) [21–24]. The study of the rate of reactions occurring during thermal processes is highly important for optimizing the process with the expected effect. The thermal characteristics of the tested waste provide information on the validity of the process. The analyses yielded the organic matter decomposition reaction rate constant and activation energy, and identified the thermal balance of phase transformations, during the thermal degradation of individual wastes. The analyses provided information on the energy intensity of the low-temperature pyrolysis of components of MSW.

## 2. Materials and Methods

### 2.1. Sample Collection and Preparation

In this research, 12 organic waste materials were chosen to represent the most abundant waste that can be found in the municipal solid waste stream. Organic waste materials were divided into six basic groups as follows: paper, cardboard, textiles, plastics, hygiene waste, and biodegradable waste. For each group, two waste materials were selected to be studied. Research materials were: (i) paper (receipts, cotton wool); (ii) cardboard (cardboard, egg carton); (iii) textiles (cotton, leather); (iv) plastics (polyethylene (PET), polyurethane (PU)); (v) hygiene waste (diapers, leno); and (vi) biodegradable waste (chicken meat, potato peel). The materials were dried in a laboratory dryer (WAMED, KBC-65W, Warsaw, Poland), and then ground using a knife mill (Testchem, LMN-100, Pszów, Poland) to a size less than 0.425 mm to ensure homogeneity. Milled materials were stored in plastic bags at room temperature.

### 2.2. Proximate Analysis

The materials were subjected to proximate analysis. The moisture content (*MC*) was determined using a laboratory dryer (WAMED, KBC-65W, Warsaw, Poland) by drying at 105 °C for 24 h. The volatile matter (*VM*) was determined using thermogravimetry equipment consisting of a laboratory balance (RADWAG, PS 750.3Y, Warsaw, Poland) coupled with a tubular furnace (Czylok, RST 40 × 200/100, Jastrzębie-Zdrój, Poland) by the TGA method [25]. *VM* was determined by pyrolysis at 950 °C for 7 min. The ash content (*AC*) was determined using a muffle furnace (Snol 8.1/1100, Utena, Lithuania) according to the procedure given by Syguła et al. [26] by sample combustion at 815 °C for 3 h. Then, the fixed carbon (*FC*) was calculated according to J.G. Speight [27] following Equation (1). Additionally, volatile solids (*VS*) were determined using a muffle furnace (Snol 8.1/1100, Utena, Lithuania) according to Randazzo et al. [28]. vs. was determined

by sample combustion at 550 for 3 h. vs. are also known as organic matter content or loss on ignition. Next, a high heating value was determined using a calorimeter (IKA® Werke GmbH, C200, Staufen, Germany) according to Świechowski et al. [29,30].

$$FC = 100\% - VM - A \tag{1}$$

where: *FC*—fixed carbon content at dry basis, %, *VM*—volatile matter content at dry basis, % *AC*—ash content at dry basis, %.

### 2.3. Thermogravimetric Analysis (TGA)

Thermogravimetric analyses were carried out using thermogravimetry equipment consisting of a laboratory balance (RADWAG, PS 750.3Y, Warsaw, Poland) coupled with a tubular furnace (Czylok, RST 40 × 200/100, Jastrzębie-Zdrój, Poland). The materials were dried before analysis. For each analysis, a 1.5 g sample was used. First, the sample was placed in a steel crucible. Next, the crucible was placed in the furnace and the furnace was filled with $CO_2$ gas. $CO_2$ was delivered to the center of the furnace at 10 $dm^3 \cdot h^{-1}$ to facilitate an inert atmosphere. Next, a sample was heated from room temperature (~20 °C) to 850 °C at a 5 °C·min$^{-1}$ heating rate. For each sample, three replications were undertaken. The mass and temperature were recorded with a 1 s interval with resolutions of 0.001 g and 1 °C, respectively. The raw TGA data were smoothed using the Loess method (smoothing parameter: span = 0.05) [31]. Then, smoothed data were used for the calculation of derivative thermogravimetry (DTG).

### 2.4. Differential Scanning Calorimetry (DSC)

Differential scanning calorimetry was performed using a differential scanning calorimeter (Mettler Toledo, DSC 822e, Warsaw, Poland). Medium pressure crucibles (120 μL) with a capacity of 5 mg were used for the analysis. The materials were heated from room temperature to 500 °C, at 5 °C·min$^{-1}$. As an inert gas, nitrogen was used at a 3.6 $dm^3 \cdot h^{-1}$ flow rate. The analysis provided information on the type of transformations occurring during the decomposition of the materials (endothermic and exothermic). Each transformation was characterized by: (i) the beginning transformation temperature; (ii) the peak transformation temperature; (iii) the ending transformation temperature, and (iv) energy needed or released for/from transformation.

### 2.5. Kinetics Analysis

The TG results were subjected to kinetic analysis of organic waste decomposition. In this study, the Coats–Redfern (CR) method was used to evaluate the kinetic triplet. The CR method is a model-free, integral method [32]. The full methodology of the CR method can be found in cited references [27–29]. The kinetic triplet for the CR method is energy activation (*Ea*), pre-exponential factor (*A*), and order of reaction (*n*). When selecting the order of reaction (*n*), the rate of reaction is given by Equation (2). After arranging and integrating, Equation (2) becomes Equation (3). Because an integral part of Equation (3) has no exact solution, asymptotic series are applied and, by neglecting higher-order terms, Equation (3) can be calculated using Equation (4) for $n \neq 1$ and Equation (5) for $n = 1$. However, because $\frac{2RT}{E_a} << 1$, Equations (4) and (5) can be simplified to the forms represented by Equations (6) and (7), respectively.

$$\frac{d\alpha}{dT} = \frac{A}{\beta} \cdot e^{\frac{E_a}{R \cdot T}} \cdot (1-\alpha)^n \tag{2}$$

$$\frac{1-(1-\alpha)^{1-n}}{1-n} = \frac{A}{\beta} \int_0^T e^{\frac{-E_a}{R \cdot T}} dT \tag{3}$$

$$ln\frac{1-(1-\alpha)^{1-n}}{T^2 \cdot (1-n)} = \ln\left\{\frac{AR}{\beta E_a} \cdot \left(1 - \frac{2RT}{E_a}\right)\right\} - \frac{E}{RT} \tag{4}$$

$$ln\frac{-ln(1-\propto)}{T^2} = ln\left\{\frac{AR}{\beta E_a}\cdot\left(1-\frac{2RT}{E_a}\right)\right\} - \frac{E}{RT} \tag{5}$$

$$ln\frac{1-(1-\propto)^{1-n}}{T^2\cdot(1-n)} = ln\left\{\frac{AR}{\beta E_a}\right\} - \frac{E}{RT} \tag{6}$$

$$ln\frac{-ln(1-\propto)}{T^2} = ln\left\{\frac{AR}{\beta E_a}\right\} - \frac{E}{RT} \tag{7}$$

where:

$\propto$—conversion (−),

$T$—absolute temperature (K),

$A$—pre-exponential factor (min$^{-1}$),

$\beta$—heating rate (°C·min$^{-1}$)

$E_a$—activation energy (kJ·mol$^{-1}$·K$^{-1}$),

$R$—universal gas constant (kJ·kmol$^{-1}$·K$^{-1}$),

In this study $n$, $Ea$, and $A$ were determined at a heating rate of 5 °C·min$^{-1}$ for $n \neq 1$. $n \neq 1$ was assumed because it resulted in a higher determination coefficient in comparison with $n = 1$. To calculate kinetic parameters, a kinetic plot between $ln(\frac{1-(1-\propto)^{1-n}}{T^2(1-n)})$ versus $\frac{1}{T}$ (for $n \neq 1$) was created. The plot provides a slope that is described by a linear equation, Equation (8).

$$y = ax + b \tag{8}$$

where:

$y$—variable equal to the value of $ln(\frac{1-(1-\propto)^{1-n}}{T^2(1-n)})$,

$a$—intercept (slope)

$x$—variable equal to the value of $\frac{1}{T}$

$b$—coefficients

Then, the activation energy and pre-exponential factor were calculated by rearranging Equations (9) and (10), respectively. Kinetic parameters were chosen for $n$ with the highest value of the determination coefficient [33].

$$a = -\frac{E_a}{R} \tag{9}$$

$$b = ln\frac{AR}{\beta E_a} \tag{10}$$

## 3. Results and Discussion

### 3.1. Proximate Analysis

The results of proximate analysis for all materials are presented in Table 1. The data indicate that the *MC* for almost all materials is less than 10%, which is important for pyrolysis because a waste with low *MC* is considered to be a good substrate for the process [34]. The lowest *MC* was found for plastics; the *MC* was 0.36% and 1.79 for PET and PU, respectively. The highest *MC* was found for biodegradable waste: the potato peel and the chicken meat had moisture greater than 50%. In all samples, the content of vs. was found to be relatively high (82.51–99.81%). The *VM* content ranged from 69.9% for potato peel to 96.7% for cotton. Moreover, the *AC* remained low, with values varying from 0.10% for PET to 14.53% for the receipts. High *VM* content indicates that tested materials are easy to ignite [35], and low ash content indicates that no problems will occur with residual disposal when materials are used as fuel. The highest *FC* content was found in leather (23.0%), and the lowest in diapers (0.4%) [34]. *FC* is the carbon remaining after devolatilization and, the higher the *FC* content, the higher the biochar yield [36], thus indicating a good substrate for pyrolysis. The highest higher heating value (*HHV*) was found for PET (42.68 MJ·kg$^{-1}$), whereas the lowest value was found for the receipts (15.01 MJ·kg$^{-1}$). An inverse proportionality was noticed between *AC* and *HHV*: lower *AC* occurs with higher *HHV* [37].

**Table 1.** Proximate analysis results of research materials.

| Material | * MC, % | ** VS, % | ** VM, % | ** FC, % | ** AC, % | HHV, MJ·kg$^{-1}$ |
|---|---|---|---|---|---|---|
| Chicken | 53.3 | 95.71 | 86.50 | 9.50 | 3.93 | 25.18 |
| Potato peel | 51.2 | 86.69 | 69.90 | 18.70 | 11.93 | 17.63 |
| Cotton | 4.56 | 99.65 | 96.70 | 2.90 | 0.33 | 22.37 |
| Leather | 9.88 | 94.60 | 73.40 | 23.00 | 3.50 | 22.84 |
| Receipt | 5.66 | 82.51 | 76.10 | 9.40 | 14.53 | 15.01 |
| Cotton wool | 5.22 | 99.41 | 95.00 | 4.50 | 0.56 | 16.27 |
| Polyethylene (PET) | 0.36 | 99.81 | 89.30 | 10.60 | 0.10 | 42.68 |
| Polyurethane (PU) | 1.79 | 91.54 | 86.10 | 8.30 | 5.57 | 15.83 |
| Diapers | 3.34 | 86.73 | 89.40 | 0.40 | 10.63 | 28.37 |
| Leno | 3.84 | 99.08 | 95.30 | 4.00 | 0.60 | 19.43 |
| Cardboard | 5.76 | 85.24 | 71.30 | 17.80 | 10.93 | 15.70 |
| Egg carton | 5.60 | 82.98 | 77.30 | 10.00 | 12.60 | 16.58 |

* as received basis ** dry basis. *MC*—moisture content, *VS*—volatile solids content, *VM*—volatile matter content, *FC*—fixed carbon content, *AC*—ash content, *HHV*—high heating value.

### 3.2. Kinetics Analysis

Based on the Coats–Redfern method, kinetic parameters, such as activation energy, pre-exponential factor, and the order of the reaction, were determined. The parameters for all samples, in addition to the linear regression coefficient, are summarized in Table 2. The average activation energy values ranged from 25.2 kJ × (mol·K)$^{-1}$ for leather to 134.5 kJ × (mol·K)$^{-1}$ for cotton, with standard deviations of 1.5 kJ × (mol·K)$^{-1}$ and 18.7 × kJ × (mol·K)$^{-1}$, respectively. Materials with lower energy activation make better feedstock for the thermochemical process because high energy activation causes the chemical reaction to slow down [38]. The pre-exponential factor is directly related to the number of times the volatiles collide, thus causing a reaction [39]. The pre-exponential factor values ranged from 2.0 min$^{-1}$ for leather to 2.3 min$^{-1}$ for cotton. This shows that, the higher the pre-exponential factor, the higher the activation energy. The determination coefficients ($R^2$) for all samples, with the exceptions of cotton wool and PET, were above 0.90, which indicates that the reaction order was correctly determined for all tested materials [32].

**Table 2.** Kinetics analysis of research materials for β = 5 K·min$^{-1}$.

| Materials | Statistics | n | E$_a$ | A | R$^2$ |
|---|---|---|---|---|---|
| | | - | kJ × (mol·K)$^{-1}$ | min$^{-1}$ | - |
| Receipt | Mean | 3.52 | 126.6 | 1.4 × 10$^9$ | 0.96 |
| | SD | 0.20 | 8,7 | 1.5 × 10$^9$ | |
| Cotton wool | Mean | 2.31 | 68,7 | 8.0 × 10$^3$ | 0.89 |
| | SD | 0.10 | 0.5 | 8.0 × 10$^2$ | |
| Cardboard | Mean | 1.46 | 32.7 | 10.0 | 0.93 |
| | SD | 0.19 | 4.6 | 8.0 | |
| Egg carton | Mean | 2.24 | 63.9 | 1.9 × 10$^4$ | 0.96 |
| | SD | 0.37 | 14.1 | 3.2 × 10$^4$ | |
| Cotton | Mean | 3.67 | 134.5 | 2.3 × 10$^{11}$ | 0.96 |
| | SD | 0.44 | 18.7 | 3.9 × 10$^{11}$ | |
| Leather | Mean | 1.25 | 25.2 | 2.0 | 0.96 |
| | SD | 0.09 | 1.5 | 1.0 | |
| Polyethylene (PET) | Mean | 2.31 | 117.0 | 4.8 × 10$^6$ | 0.87 |
| | SD | 0.01 | 1.0 | 7.2 × 10$^5$ | |
| Polyurethane (PU) | Mean | 1.60 | 51.5 | 5.3 × 10$^3$ | 0.95 |
| | SD | 0.63 | 24.3 | 9.0 × 10$^3$ | |
| Diapers | Mean | 1.41 | 62.1 | 6.5 × 10$^3$ | 0.91 |
| | SD | 0.52 | 29.8 | 6.0 × 10$^3$ | |
| Leno | Mean | 1.60 | 55.9 | 6.3 × 10$^2$ | 0.94 |
| | SD | 0.16 | 9.0 | 8.0 × 10$^2$ | |
| Chicken meat | Mean | 2.38 | 76.4 | 1.5 × 10$^4$ | 0.97 |
| | SD | 0.07 | 2.8 | 7.3 × 10$^3$ | |
| Potato peel | Mean | 2.48 | 71.7 | 7.9 × 10$^4$ | 0.97 |
| | SD | 0.42 | 10.7 | 8.7 × 10$^4$ | |

### 3.3. TG/DTG and DSC Analysis

#### 3.3.1. Paper

A decrease in mass as a function of temperature for the receipts (Figure 1a) and the cotton wool was observed (Figure 1b). The lag phase of the cotton lasted 36 min and that of the receipts lasted 50 min, at temperatures of 194 and 258 °C, respectively. The most dynamic weight decreases for cotton wool occurred between 77 and 101 min, from 396 to 513 °C, respectively, whereas that for receipts occurred between 68 and 95 min, from 350 to 480 °C, respectively. An observed decrease in mass was related to the decomposition of organic compounds, such as hemicellulose, cellulose, and lignin. The thermal degradation of these compounds occurred in the ranges of 220–320 °C, 320−400 °C, and 320−900 °C, respectively [40,41].

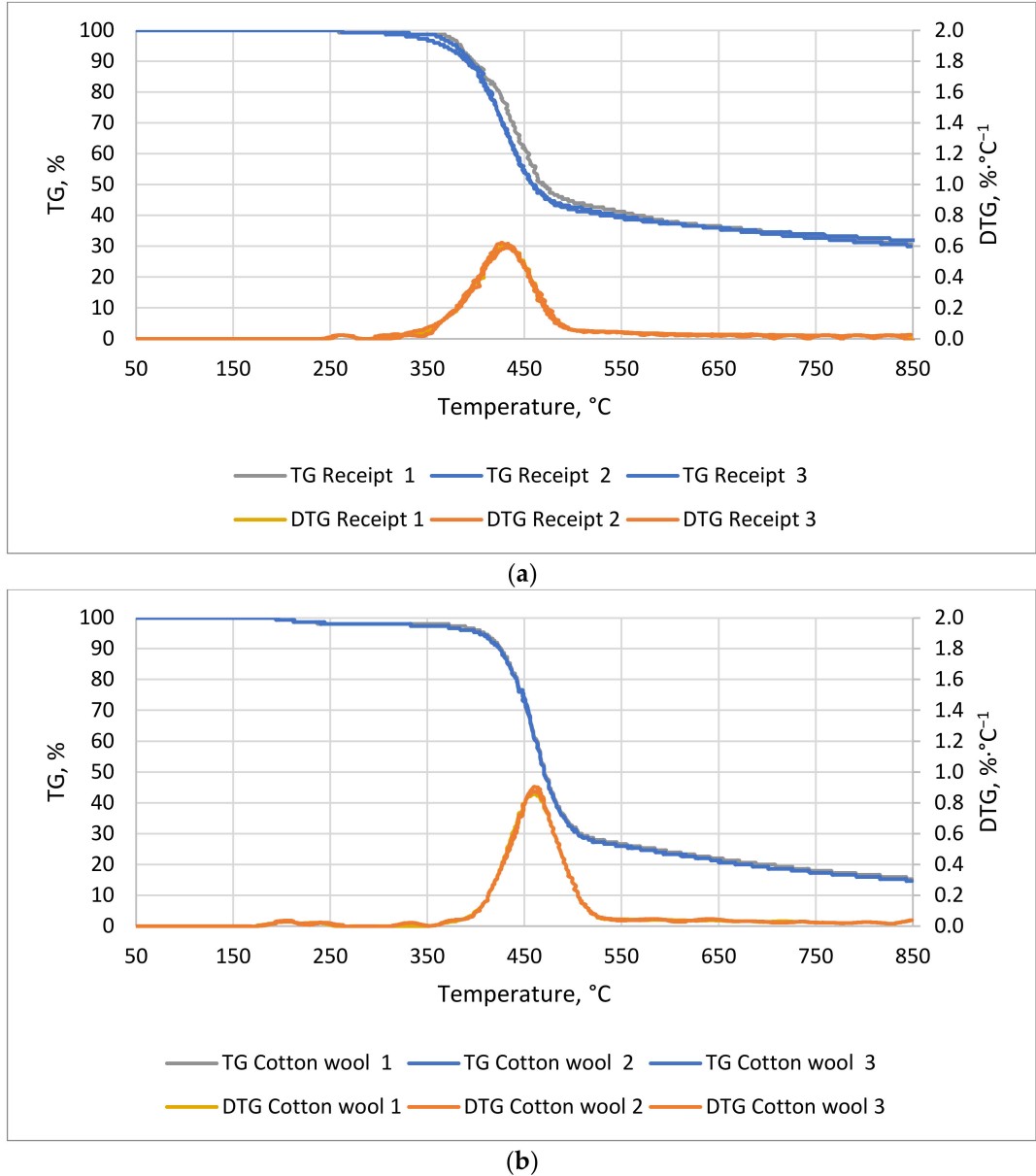

**Figure 1.** Characteristics of the TG/DTG of paper waste: (**a**) receipts; (**b**) cotton wool.

Three transformations for both receipts (Figure 2a) and cotton wool (Figure 2b) were identified during DSC tests. The first and the second were exothermic, and the third was endothermic. These transformations were also characterized by similar start and

end temperatures. The balance of these transformations was higher for the cotton wool sample. Both samples are composed mainly of cellulose. The receipts additionally include the compounds present in the carcasses. The first of the transformations, in both samples, occurred at about 200 °C. This explains the onset of hemicellulose depolymerization and, in the case of receipts, it may also be related to bisphenol A, which is present in some receipts. The boiling point of this compound is 220 °C [42]. The temperatures of the other two transformations indicate that they are associated with the decomposition of biopolymers: hemicellulose at 220−320 °C, cellulose at 320−400 °C, and lignin at 320−900 °C [43].

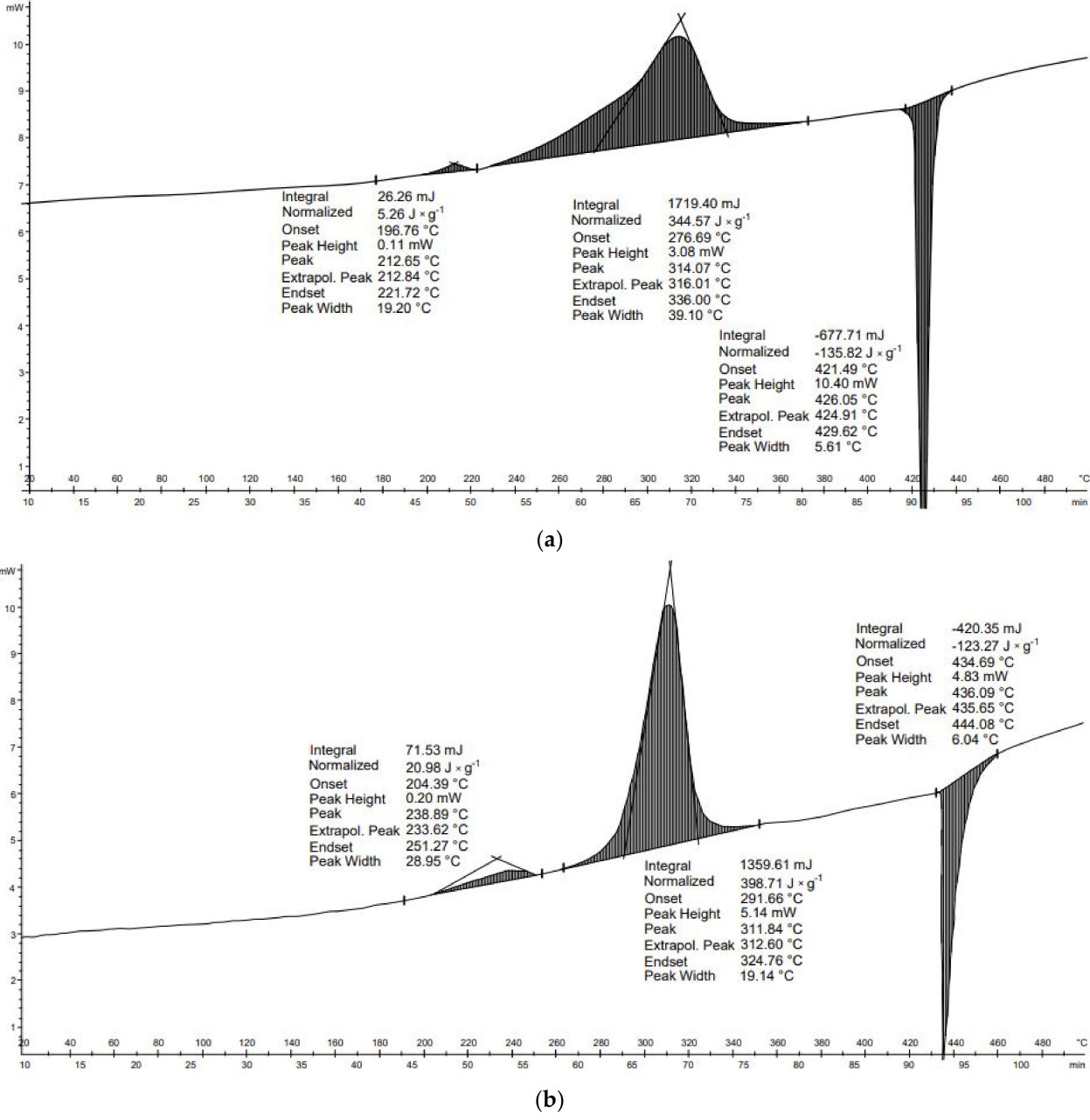

**Figure 2.** Characteristics of the DSC of paper waste: (**a**) receipts; (**b**) cotton wool.

The energy balance determines the energy of transformations occurring during the thermal conversion process of the tested materials (Table 3). Reactions are divided into exothermic and endothermic.

**Table 3.** Normalized transformation energy for successive transformations of tested paper waste and energy balance of the process.

| DSC Characteristics | Type of Material | Transformation | | | | | | | |
|---|---|---|---|---|---|---|---|---|---|
| | | 1 | 2 | 3 | 4 | 5 | 6 | 7 | |
| The temperature of transformation beginning, °C | Receipt | 196.8 | 276.7 | 421.5 | - | - | - | - | |
| The temperature of transformation peak, °C | | 212.7 | 314.1 | 426.1 | - | - | - | - | |
| The temperature of transformation end, °C | | 221.7 | 336.0 | 429.6 | - | - | - | - | |
| Energy balance, $J \times g^{-1}$ | | 5.3 | 344.6 | −135.9 | - | - | - | - | 214.0 |
| The temperature of transformation beginning, °C | Cotton wool | 204.4 | 291.7 | 434.7 | - | - | - | - | |
| The temperature of transformation peak, °C | | 238.9 | 311.8 | 436.1 | - | - | - | - | |
| The temperature of transformation end, °C | | 251.3 | 324.8 | 444.1 | - | - | - | - | |
| Energy balance, $J \times g^{-1}$ | | 21.0 | 398.7 | −123.3 | - | - | - | - | 296.4 |

Three transformations occurred in the thermal analysis of the receipts. The first and second transformations were exothermic and released 5.3 and 344.6 $J \times g^{-1}$ of energy, respectively. The third transformation was endothermic and absorbed 135.9 $J \times g^{-1}$. As a result, the energy balance of thermal transformation for receipts was positive (214.0 $J \times g^{-1}$). This means that more energy was released than absorbed during transformations. In the case of cotton wool, the first transformation was exothermic (21.0 $J \times g^{-1}$), the second transformation exothermic (399.0 $J \times g^{-1}$), and the final transformation endothermic (−123.3 $J \times g^{-1}$). The energy balance of the thermal conversion of this material was positive (296.4 $J \times g^{-1}$).

For both materials, DSC results show that an endothermal reaction occurred above 400 °C (Figure 2). This suggests that temperatures above 400 °C should be avoided to decrease the energy needs for the pyrolysis process of paper wastes.

Paper wastes are a mixture of materials such as cellulose, hemicellulose, and lignin, and other chemical additives that might affect results obtained from DSC results. In the work of Stępień et al. [14], various elements of refuse-derived fuel (RDF) were tested by TGA/DSC analyses to determine the energy needed for the torrefaction process. During torrefaction, the material is heated from room temperature to 300 °C. According to Stępień et al. [14], paper waste has the highest energy demand of 1061 $J \times g^{-1}$. These findings confirm that high temperatures should be avoided to ensure a low level of required process energy.

### 3.3.2. Cardboard

In the case of cardboard (Figure 3a), the first mass decrease occurred around 28 min at 150 °C and the second occurred at 73 min at 375 °C. The first lasted about 18 min and the second 28 min. The mass decreases ended at 239 and 510 °C, respectively. For the egg cardboard (Figure 3b), the weight decrease occurred at about 64 min at 330 °C and lasted for 32 min. The decrease in weight for cardboard and egg cartons was mainly related to the decomposition of the pulp, which consists of biopolymers such as cellulose, hemicellulose, and lignin. The decrease in mass began between 200 and 300 °C and was associated with the decomposition of the aforementioned biopolymers [44].

DSC characteristics of cardboard (Figure 4a) show four transformations—the first, second, and third are exothermic, whereas the fourth is endothermic. Cardboard is a cellulosic material, so both its thermal decomposition and phase transformations should be similar to those of leno, cotton, and paper waste. This is also the case for the egg cardboard sample (Figure 4b). Unprocessed cardboard wastes contain approximately 32% hemicellulose, 6% lignin, 48% cellulose, and 14% inorganic matter [40]. The similarity of these materials is that the first transformations are exothermic, whereas the latter are endothermic. There processes are associated with the breakdown of hemicellulose, cellulose, and lignin. They are therefore distributed between 220 and 900 °C.

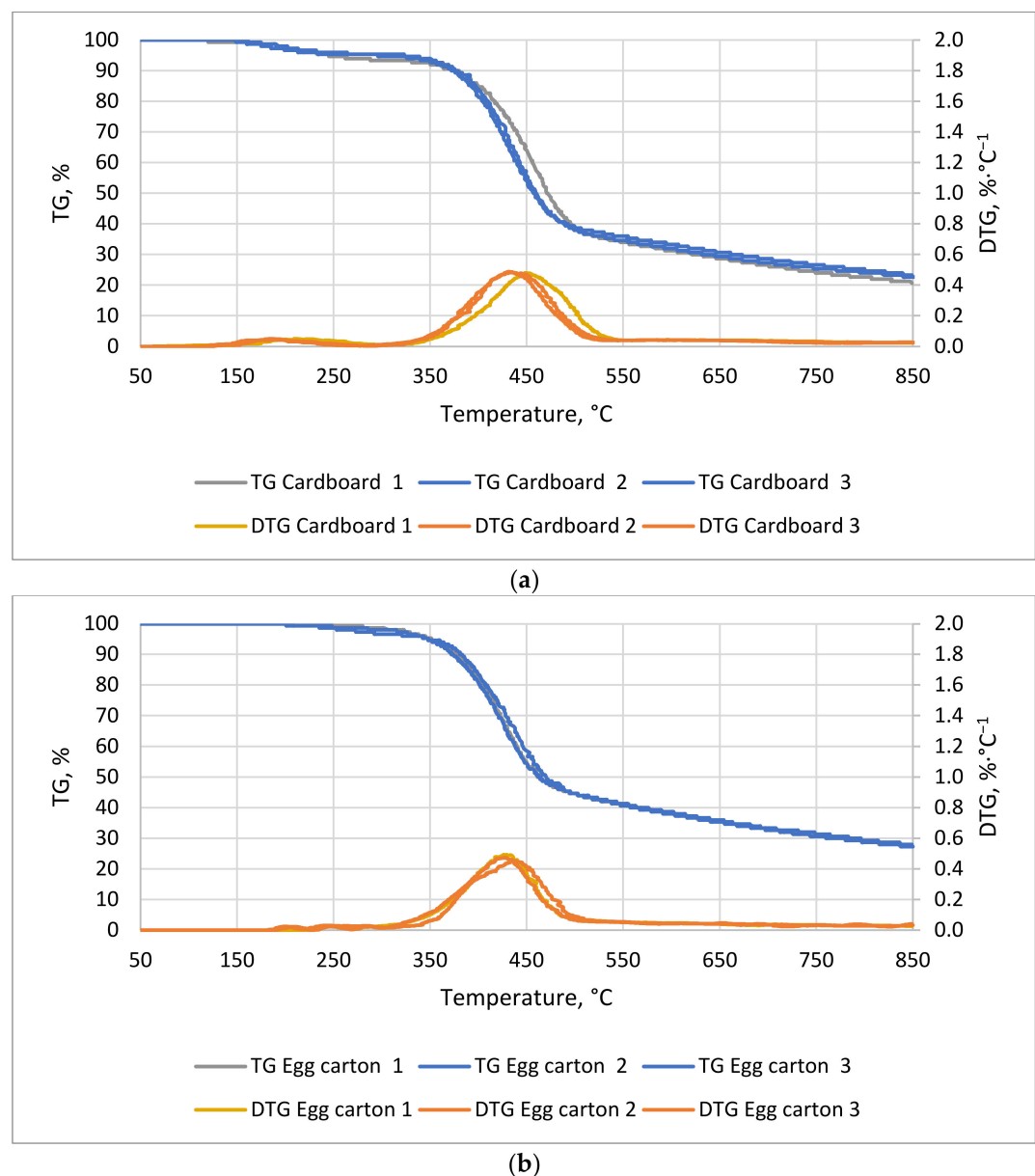

**Figure 3.** Characteristics of the TG/DTG of cardboard waste: (**a**) cardboard; (**b**) egg cardboard.

Four transformations occurred in the thermal analysis of cardboard. The first three transformations were exothermic, and led to the release of heat of 12.1, 357.0, and 10.4 J × g$^{-1}$, respectively (Table 4). The final transformation was endothermic: $-188.8$ J × g$^{-1}$. The energy balance of the transformation of the cardboard was positive (190.8 J × g$^{-1}$). In the case of egg cardboard, three transformations occurred. The first and second transformations were exothermic: 67.0 J × g$^{-1}$ and 159.5 J × g$^{-1}$, respectively. The third, however, was endothermic ($-132.4$ J × g$^{-1}$). As a result, the energy balance of the thermal transformations of egg cardboard was exothermic (94.0 J × g$^{-1}$). For both materials, DSC results show that the biggest exothermal reactions took place at 220–340 °C, and endothermal reactions occurred above 420 °C (Figure 2). This suggests that temperatures above 420 °C should be avoided and process temperatures should be maintained in the range of 220–340 °C to reduce the energy needs for the pyrolysis process of cardboard wastes.

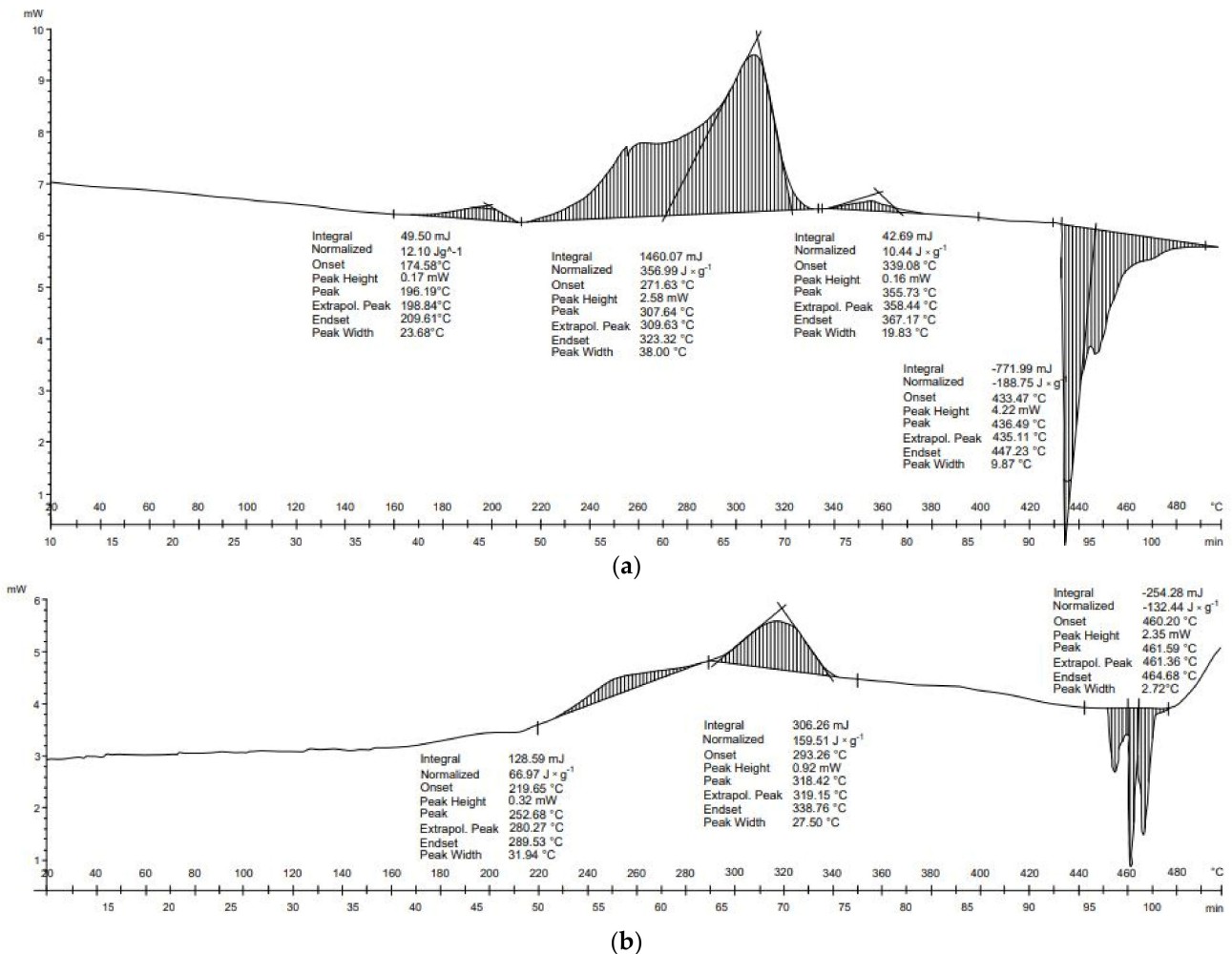

**Figure 4.** Characteristics of the DSC of cardboard waste: (**a**) cardboard; (**b**) egg cardboard.

**Table 4.** Normalized transformation energy for successive transformations of tested cardboard waste and energy balance of the process.

| DSC Characteristics | Type of Material | Transformation | | | | | | | |
|---|---|---|---|---|---|---|---|---|---|
| | | 1 | 2 | 3 | 4 | 5 | 6 | 7 | |
| The temperature of transformation beginning, °C | | 174.6 | 271.6 | 339.1 | 433.5 | - | - | - | |
| The temperature of transformation peak, °C | Cardboard | 196.2 | 307.6 | 355.7 | 436.5 | - | - | - | |
| The temperature of transformation end, °C | | 209.6 | 323.3 | 367.2 | 447.2 | - | - | - | |
| Energy balance, $J \times g^{-1}$ | | 12.1 | 357.0 | 10.4 | −188.8 | - | - | - | 190.8 |
| The temperature of transformation beginning, °C | | 219.7 | 293.3 | 460.2 | - | - | - | - | |
| The temperature of transformation peak, °C | Egg carton | 252.7 | 318.4 | 461.6 | - | - | - | - | |
| The temperature of transformation end, °C | | 289.5 | 338.8 | 464.7 | - | - | - | - | |
| Energy balance, $J \times g^{-1}$ | | 67.0 | 159.5 | −132.4 | - | - | - | - | 94.0 |

In the work of Stępień et al. [45], the energy required for cardboard torrefaction was determined. The total energy needs were 1312 $J \times g^{-1}$. This value was higher than for paper wastes, and these results are consistent with those obtained in the current study. Here, for cardboard waste, transformations showed a lower positive energy balance than for paper wastes. This indicates that more energy is needed for endothermic transformations or exothermic transformations, providing less energy than in the case of paper wastes. The DSC investigations performed in the current study correspond to low-temperature pyrolysis in which the carton was characterized by relatively low energy of transformation for the carton of 190.8 $J \times g^{-1}$, and for egg cartons of 94.0 $J \times g^{-1}$; however, these transfor-

mations concern only the transformations of the sample, not the whole energy demand of the process [45].

### 3.3.3. Textiles

The TG/DTG curves for natural leather and cotton textile are shown in Figure 5a,b. For cotton textile, the first stage of the process, i.e., the lag phase, took longer than for natural leather, 53 and 23 min, respectively. The weight of the leather sample showed two distinct reductions: the first was between 23 and 53 min at 130–274 °C, and the second was between 70 and 116 min at 360–585 °C. The final weight drop for natural leather occurred at 167 min and for cotton at 170 min of measurement.

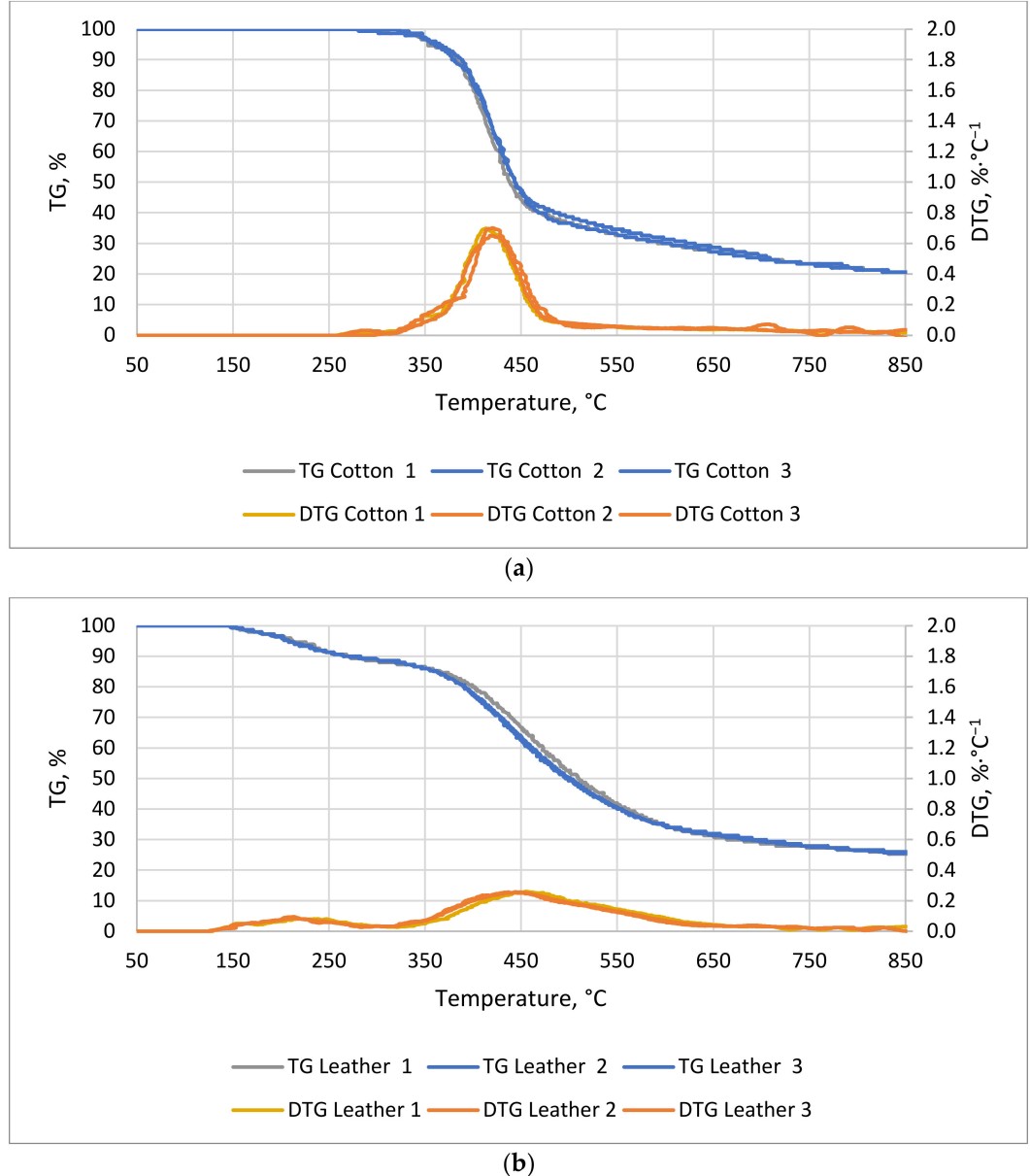

**Figure 5.** Characteristics of the TG/DTG of textile waste: (**a**) cotton textile; (**b**) leather.

The DSC characteristics of the cotton textile are shown in Figure 6a. Five transformations are visible, one exothermic and four endothermic. The first transformation started at about 251 °C. At this temperature, hemicellulose and cellulose, which are the main components of cotton, are broken down. In addition, further transformations that originate at about 310 to 415 °C, respectively, are associated with these biopolymers [46].

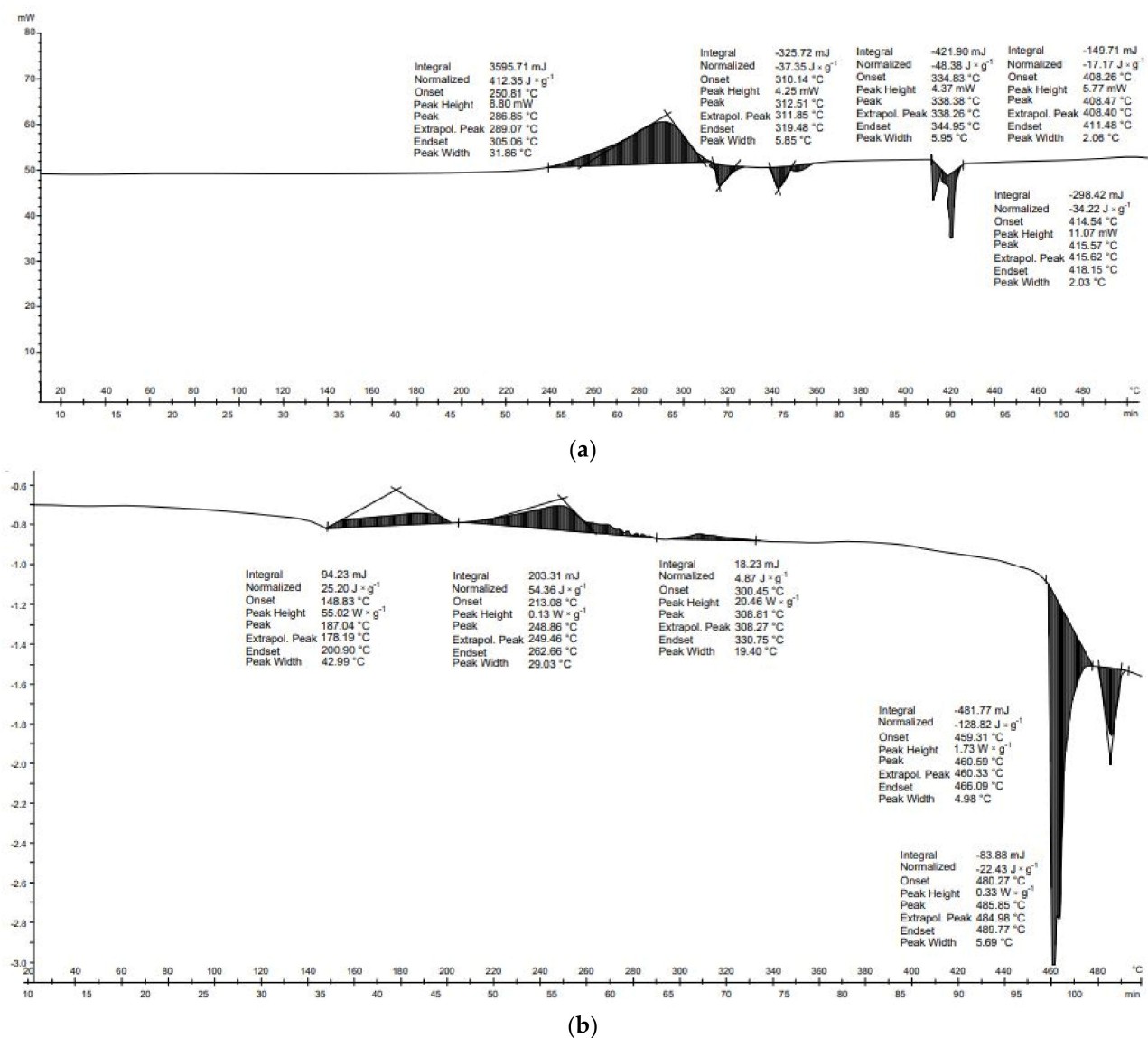

**Figure 6.** Characteristics of the DSC of textile waste: (**a**) cotton textile; (**b**) leather.

Comparing the DTG curves of leno (Figure 9b) and cotton (Figure 5a), only one exothermic transformation occurred for these materials. The transformations of the leno sample started at lower temperatures and ended at higher temperatures compared to the cotton sample. The two materials tested differ in the number of agents added; in the case of leno, only hydrogen peroxide is added, whereas in the case of cotton several different compounds associated with fabric dyeing may be added. According to the characteristics of textile dyeing technology in the EU, in addition to dyes, these agents can include sodium hydroxide, sodium carbonate, sodium chloride, or sodium sulfate. These melt at 318–323, 854, 801, and 884 °C, respectively [47]. This shows that sodium hydroxide may have affected the exothermic peak that occurred between 310 and 319 °C. As in the case of the leno between 380 and 600 °C, a main pyrolysis process occurs that produces L-glucose and various combustible leno products. In addition, several dyes of various origins can affect the thermal decomposition process and phase transformations of the cotton sample. These dyes may include canthaxanthin, which has a melting point of 217 °C; vitamin B2, which has a melting point of 290 °C; sodium chlorate, which has a melting point of 247 °C and a boiling point 300°C; and para-aminophenol, which has a melting point of 186 °C and a boiling point of 284 °C.

Figure 6b shows the DSC characteristics for natural leather. The leather is composed of water, proteins, fats, and mineral salts. Because the sample was dry and protein denaturation occurs at temperatures below 100 °C, the peaks in Figure 6b cannot be assumed to be associated with water removal and protein transformation. It is more likely that they are related to fat breakdown and protein product denaturation. In addition, other additives added during leather production, e.g., dyes, may react during the leather degradation. As a result, unknown reactions could appear. Nevertheless, the first three transformations were exothermic and the next two were endothermic.

Five transformations occurred in the thermal analysis of cotton textile. The first transformation was exothermic ($412.4 \text{ J} \times \text{g}^{-1}$), and the following transformations were endothermic ($-37.4$, $-48.4$, $-17.2$, and $-34.2 \text{ J} \times \text{g}^{-1}$, respectively). The overall energy balance of the thermal transformations of the cotton textile was positive ($275.2 \text{ J} \times \text{g}^{-1}$) (Table 5). Five thermal transformations occurred during the process of thermal leather analysis. The first three transformations were exothermic ($25.2$, $54.4$, and $4.9 \text{ J} \times \text{g}^{-1}$, respectively). The fourth and fifth transformations were endothermic ($-128.9 \text{ J} \times \text{g}^{-1}$ and $-22.4 \text{ J} \times \text{g}^{-1}$, respectively). The overall energy balance of the thermal transformations of the leather was negative ($-66.8 \text{ J} \times \text{g}^{-1}$) (Table 5). Results showed that temperatures higher than 440 °C should be avoided to reduce the energy needed for leather because, above this temperature, two significant endothermic reactions take place that require a large amount of energy, and this energy cannot be supplied by previous exothermic reactions. This problem does not exist in the case of cotton textile because the total energy balance of the reaction is positive.

**Table 5.** Normalized transformation energy for successive transformations of tested textile waste and energy balance of the process.

| DSC Characteristics | Type of Material | Transformation | | | | | | | |
|---|---|---|---|---|---|---|---|---|---|
| | | 1 | 2 | 3 | 4 | 5 | 6 | 7 | |
| The temperature of transformation beginning, °C | Cotton textile | 250.8 | 310.1 | 334.8 | 408.3 | 414.5 | - | - | |
| The temperature of transformation peak, °C | | 286.9 | 312.6 | 338.4 | 408.5 | 415.6 | - | - | |
| The temperature of transformation end, °C | | 305.1 | 319.5 | 345.0 | 411.5 | 418.2 | - | - | |
| Energy balance, $\text{J} \times \text{g}^{-1}$ | | 412.4 | −37.4 | −48.4 | −17.2 | −34.2 | - | - | 275.2 |
| The temperature of transformation beginning, °C | Leather | 148.8 | 213.1 | 300.5 | 459.3 | 480.3 | - | - | |
| The temperature of transformation peak, °C | | 187.0 | 248.9 | 308.9 | 460.6 | 485.9 | - | - | |
| The temperature of transformation end, °C | | 200.1 | 262.7 | 330.8 | 466.1 | 489.8 | - | - | |
| Energy balance, $\text{J} \times \text{g}^{-1}$ | | 25.2 | 54.4 | 4.9 | −128.9 | −22.4 | - | - | −66.8 |

Nevertheless, these findings might not be representative of each textile's wastes. A study by Hassabo et al. [48] was based on the characterization of cotton by TGA and DSC to assess thermal stability. The cotton waste came from the recycling of textile waste. The study confirmed that, as the cotton particle size decreased, its thermal stability increased. Stępień et al. [45] conducted TGA and DSC analyses to evaluate the energy intensity of using a torrefaction process to produce carbonized alternative fuel. His results showed that, in the case of textiles, dyes contained in fabrics can alter the course of phase transformations. In both the current study and that of Stępień et al. [45], there was an exothermic transformation at 300 °C, and the value of transformation in Stępień et al. [45] ($423 \text{ J} \cdot \text{g}^{-1}$) was similar to the value of $412 \text{ J} \cdot \text{g}^{-1}$ obtained in this study. In contrast, natural leather processed up to 300 °C by Stępień et al. [45] was also characterized by three exothermal transformations, for which the overall energy balance was $476 \text{ J} \cdot \text{g}^{-1}$, whereas in the present study it was only $84.5 \text{ J} \cdot \text{g}^{-1}$. These differences may result from the choice of material with different properties, size distribution, and additives [14,45].

### 3.3.4. Plastics

Figure 7a,b show the TG/DTG curves of PU and PET. Polyurethane began to decompose much faster than PET (Figure 7a). The first noticeable mass decrease in PU occurred at about 58 min of the process at a temperature of about 295 °C. For the PET sample,

the first noticeable decrease in mass occurred around 95 min at 480 °C. The time at which the greatest decrease in material weight occurred was 24 and 57 min for PET and PU, respectively. The thermal decomposition time of the PET sample started at a significantly higher temperature and was less than half that of the PU sample. The conclusion of weight loss occurred at 119 min for PET and 114 min for PU at 595 and 572 °C, respectively. The first slight weight loss may be due to the phase change of compounds that are added to the plastics during their production, or the phase change associated with the transition from the solid to the plastic phase. A weight change in the PU sample was observed from about 120 °C but, up to 300 °C, the decreases were very small (Figure 7b). These decreases can be combined with the transformations of isocyanates, which have a boiling point of about 208 °C. The most dynamic mass decrease was observed in the temperature range from 320 to 600 °C. The decrease in weight of this material is mainly related to the decomposition of polyols, which is the second of the two main components of polyurethane [49].

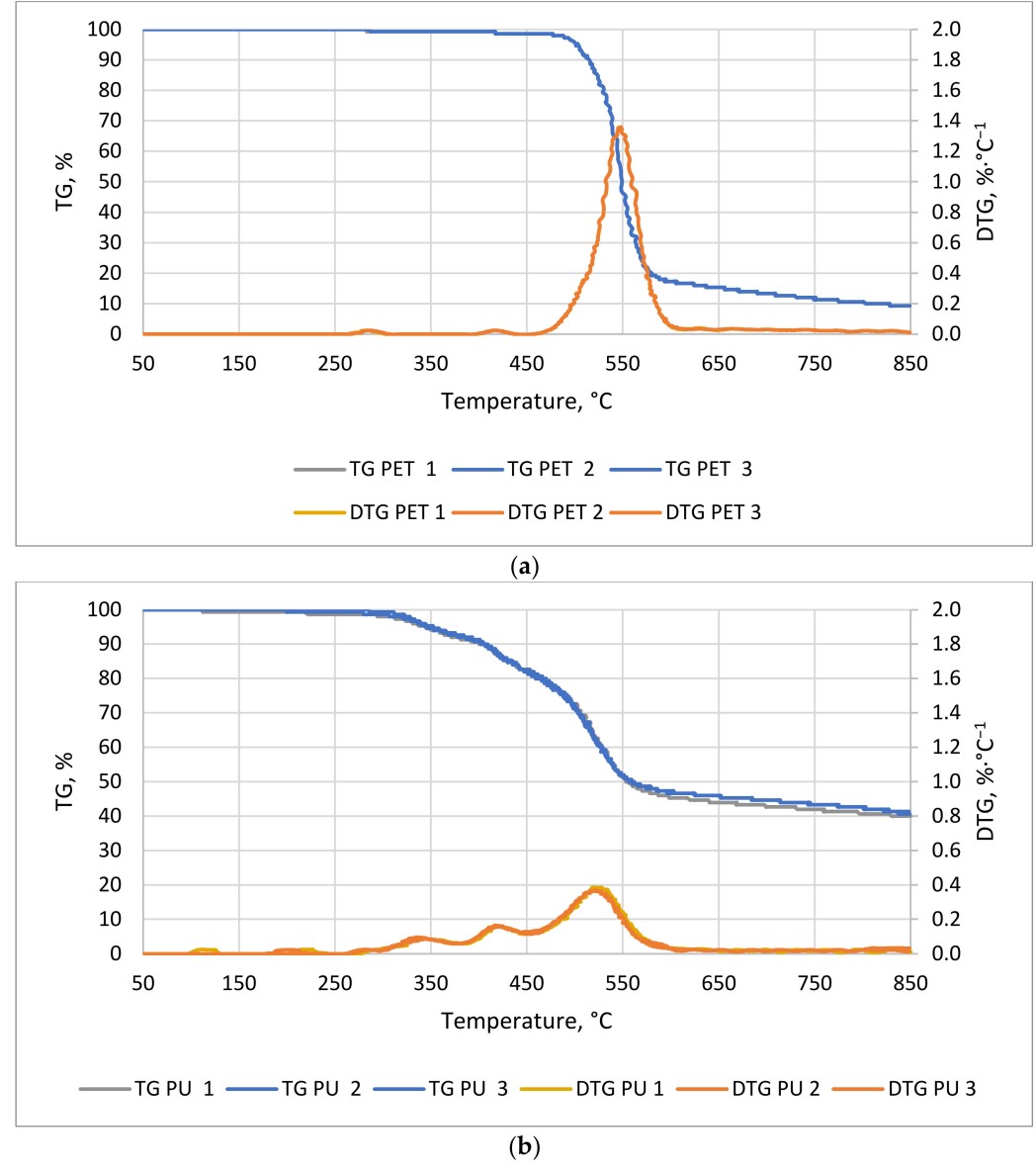

**Figure 7.** Characteristics of the TG/DTG of plastic waste: (**a**) PET; (**b**) PU.

For PU, seven transformations were recorded during DSC analysis (Figure 8b). Polyurethane materials lose their thermal strength between 150 and 200 °C, decomposing into products of different compositions and toxicity [50]. The first transformation occurred

between 64 and 161 °C. Due to the large range of temperatures, it is not possible to identify which compounds have been transformed; it may be mannitol or erythritol, which have melting points of 168 and 120 °C, respectively. Another transformation from 191 to 216 °C can be associated with the boiling of ethyl glycol—198 °C; melting of perseitol—about 185 °C; boiling of xylitol—216 °C; or boiling of isocyanate—208 °C [51,52]. The third transformation started at 219 °C and ended at 239 °C; this is associated with the boiling of polycarbonate, which has a melting point of 230 °C, or the melting of inositol (222–227 °C) [53]. Another transformation that can be related to the phase transformations of polyols starts at 289 °C and ends at 325 °C. This temperature range includes the boiling points of glycerol—209 °C, inositol—291 °C, perseitol—296 °C, and erythritol—about 330 °C [54]. The transformations described thus far are not responsible for the significant mass loss that occurred between 300 and 550 °C (Figure 8). The final transformation (374–460 °C), which is the most exothermal, is related to the critical point of glycol, whose critical point temperature is 446 °C.

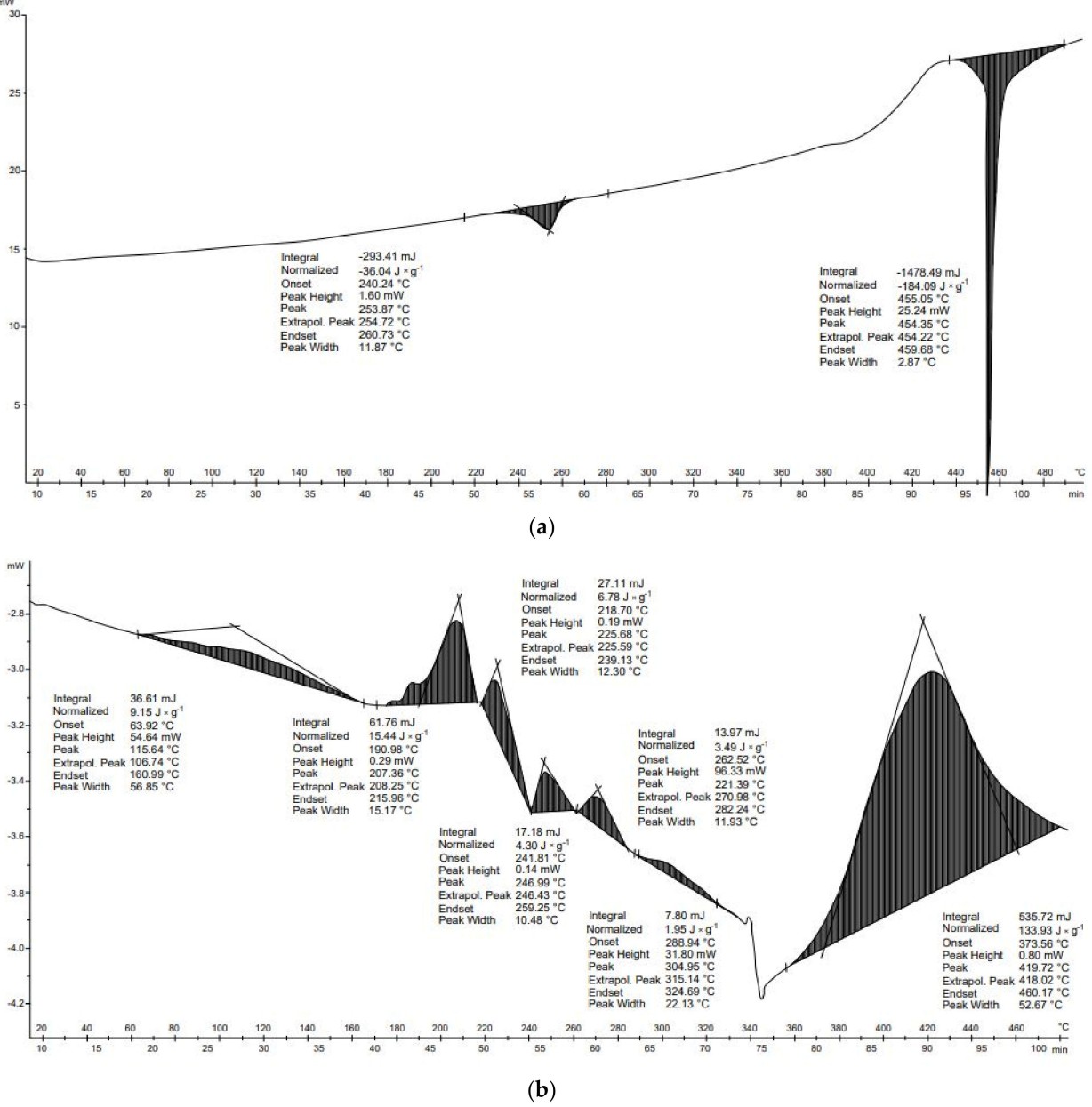

**Figure 8.** Characteristics of the DSC of plastic waste: (**a**) PET; (**b**) PU.

The DSC curve of PET material shows that the first transformation occurred between 240 and 261 °C (Figure 8b). At the same temperature, however, there was no change in mass during DTG/TG analysis. In the case of PET, the first decrease in mass occurred at around 300 °C and the second at around 400 °C, but they were small, at the 0.01 g level. A noticeable decrease in mass did not occur until about 480 °C and continued until about 600 °C. Two transformations occurred in the thermal analysis of PET. The first transformation was endothermic ($-36.0$ J $\times$ g$^{-1}$) and the second transformation was endothermic ($-184.1$ J $\times$ g$^{-1}$). The energy balance of the thermal transformation of the PET was negative ($-220.2$ J $\times$ g$^{-1}$) (Table 6). In the case of PU, seven thermal transformations occurred during thermal analysis. Each of these was exothermic. These transformations produced 9.2, 15.4, 6.8, 4.3, 3.5, 2.0, and 133.9 J $\times$ g$^{-1}$, respectively. The transformations energy balance of the PU material was positive (175.0 J $\times$ g$^{-1}$) (Table 6).

**Table 6.** Normalized transformation energy for successive transformations of tested plastic waste and energy balance of the process.

| DSC Characteristics | Type of Material | Transformation | | | | | | | |
|---|---|---|---|---|---|---|---|---|---|
| | | 1 | 2 | 3 | 4 | 5 | 6 | 7 | |
| The temperature of transformation beginning, °C | PET | 240.2 | 455.1 | - | - | - | - | - | |
| The temperature of transformation peak, °C | | 253.9 | 454.4 | - | - | - | - | - | |
| The temperature of transformation end, °C | | 260.7 | 459.7 | - | - | - | - | - | |
| Energy balance, J $\times$ g$^{-1}$ | | $-36.0$ | $-184.1$ | - | - | - | - | - | $-220.2$ |
| The temperature of transformation beginning, °C | PU | 63.9 | 191.0 | 218.7 | 241.8 | 262.5 | 288.9 | 373.6 | |
| The temperature of transformation peak, °C | | 115.6 | 207.4 | 225.7 | 247.0 | 271.4 | 305.0 | 419.7 | |
| The temperature of transformation end, °C | | 161.0 | 216.0 | 239.1 | 259.3 | 282.2 | 324.7 | 460.2 | |
| Energy balance, J $\times$ g$^{-1}$ | | 9.2 | 15.4 | 6.8 | 4.3 | 3.5 | 2.0 | 133.9 | 175.0 |

TGA and DSC analyses of plastics are mainly carried out to verify the presence of impurities in these materials. Nevertheless, studies of Chhabra et al. [55] also provided information about the energy consumption for pyrolysis of different plastic materials. Results showed that the required energy depends on the plastic type and differs for each transformation effect. These conclusions are consistent with those obtained in this study. Nevertheless, in the study of Chhabra et al. [55], PET transformations had enthalpy of about 768 J·g$^{-1}$, whereas in this study this energy was $-220.2$ J $\times$ g$^{-1}$, which may result from different measurement methods (different heating rate, sample mass, final temperature) and impurities in samples [55].

### 3.3.5. Hygiene Waste

The TG/DTG curves of hygiene waste are shown in Figure 9a for diapers and Figure 9b for leno. The first noticeable mass decrease for both materials was observed around 70 min at 355 °C for pampers and 345 °C for leno. The most dynamic mass decrease lasted 51 and 48 min, and ended at 620 and 571 °C, respectively. The weight loss of diapers (Figure 9a) is related to the distribution of superabsorbent (superabsorbent polymers) and cellulose. These account for 33% and 24% of the weight of disposable diapers, respectively [56]. In the case of leno (Figure 9b), the decrease in mass is mainly due to the decomposition of cotton, i.e., cellulose, from which the dressing leno is made. For both materials, the most dynamic weight loss began at 350 °C and lasted until 600 °C.

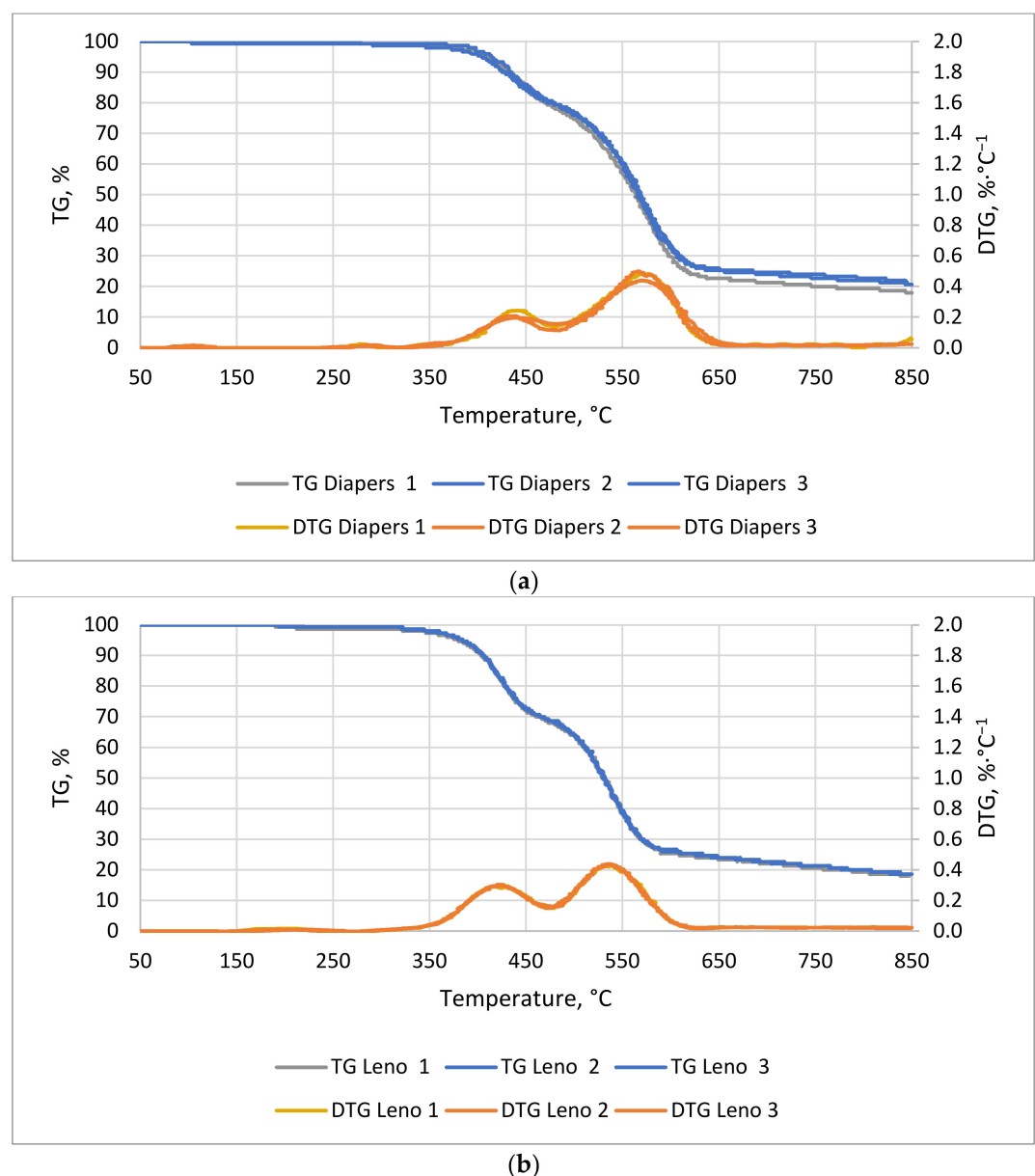

**Figure 9.** Characteristics of the TG/DTG of hygiene waste: (**a**) diapers; (**b**) leno.

The DSC curve for the diapers is shown in Figure 10a. Six transformations—four endothermal and two exothermal—were observed. The first two transformations occurred in the temperature ranges of 124–131 °C and 156–172 °C. These transformations can be related to the melting point of PEHD plastic (125 °C), which represents about 2.2% of the diaper weight, and the melting of polypropylene (PP). PP constitutes 5.8% of the disposable diaper, and its melting occurs between 160 and 170 °C [57]. The next two exothermal transformations are related to the decomposition of cellulose pulp, which constitutes about 24%, and the decomposition of superabsorbents, which constitute about 33% of the pampers weight. Cellulose pulp, like biopolymers, decomposes between 220 and 400 °C (hemicellulose and cellulose). This form of cellulose, due to prior processing, can begin to decompose at lower temperatures than the cellulose contained in cardboard or plants. The third transformation results from the decomposition of hemicellulose, whereas the fourth results from the decomposition of cellulose. The latter transformations are endothermic. They occur at higher temperatures than the others; between the end of the fourth transformation and the beginning of the fifth, there is a difference of about 115 °C. These transformations may

result from the decomposition of lignin, but may also be associated with the decomposition of substances formed during earlier transformations [40].

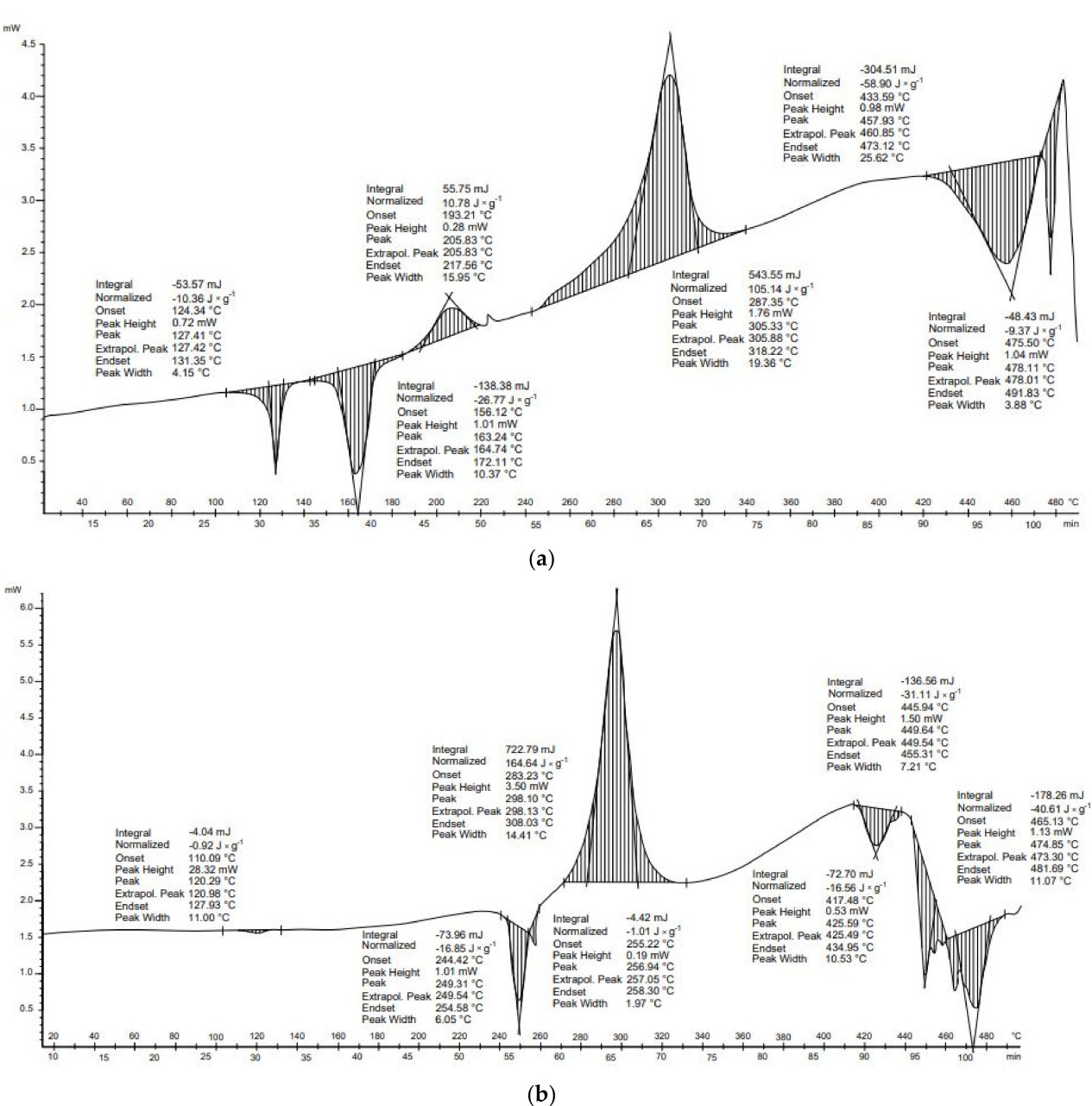

**Figure 10.** Characteristics of the DSC of hygiene waste: (**a**) diapers; (**b**) leno.

Figure 10b shows the DSC analysis of the dressing leno. According to the manufacturer's description, it is made of 100% cotton and is bleached with hydrogen peroxide. The thermal decomposition of this material is related mainly to the decomposition of the cellulose fibers and to the reactions undergone by hydrogen peroxide. The first endothermic transformation originates at 110 °C and is related to the breakdown of cotton fibers, which occurs between 110 and 150 °C. This transformation can also be associated with the boiling of hydrogen peroxide, whose boiling point is 150 °C. During these transformations, primarily physical changes occur in the cotton fibers. The subsequent transformations can be related to the decomposition of the main component of the leno, namely cellulose and hemicellulose, whose decomposition occurs from 220 to 400°C. The largest and most dynamic mass loss occurred between 380 and 600 °C. The main pyrolysis process also

occurs at these temperatures. In this temperature range, L-glucose and various combustible gases are formed. The sixth and seventh transformations, which are dehydration and carbonization processes, occur above 430 °C, and compete with the production of L-glucose. A decarboxylation process also takes place, resulting in the release of carbon dioxide. With increasing temperature, the carbon content increases. The pyrolysis of cotton products produces compounds such as water, carbon monoxide, carbon dioxide, alcohols, aldehydes, ketones, esters, ethers, and benzene. However, these compounds are only formed by the pyrolysis of non-combustible cotton.

Six transformations occurred in the process of thermal analysis of the diapers. The first transformation was endothermic ($-10.4$ J $\times$ g$^{-1}$), the second transformation was endothermic ($-26.8$ J $\times$ g$^{-1}$), the third transformation was exothermic ($10.8$ J $\times$ g$^{-1}$), the fourth transformation was exothermic ($105.1$ J $\times$ g$^{-1}$), the fifth transformation was endothermic ($-58.9$ J $\times$ g$^{-1}$), and the final transformation was endothermic ($-9.3$ J $\times$ g$^{-1}$). The energy balance of the thermal transformation of the diapers was positive ($10.5$ J $\times$ g$^{-1}$) (Table 7). In the case of leno, seven thermal transformations occurred. The first transformation was endothermic ($-0.92$ J $\times$ g$^{-1}$), the second transformation was endothermic ($-16.9$ J $\times$ g$^{-1}$), the third transformation was endothermic ($-1.0$ J $\times$ g$^{-1}$), the fourth transformation was exothermic ($164.6$ J $\times$ g$^{-1}$), the fifth transformation was exothermic ($-72.7$ J $\times$ g$^{-1}$), the sixth transformation was endothermic ($-31.1$ J $\times$ g$^{-1}$), and the final transformation was endothermic ($-40.3$ J $\times$ g$^{-1}$). The energy balance of the thermal transformations of the material was positive ($1.4$ J $\times$ g$^{-1}$) (Table 7). These results suggest that, to reduce energy needs for pyrolysis of hygiene waste, the process temperature should be no higher than 400–420 °C. This is because endothermic reactions occur above these temperatures.

**Table 7.** Normalized transformation energy for successive transformations of tested hygiene waste and energy balance of the process.

| DSC Characteristics | Type of Material | Transformation | | | | | | | |
|---|---|---|---|---|---|---|---|---|---|
| | | 1 | 2 | 3 | 4 | 5 | 6 | 7 | |
| The temperature of transformation beginning, °C | | 124.4 | 156.1 | 193.2 | 287.4 | 433.6 | 475.5 | - | |
| The temperature of transformation peak, °C | Diapers | 127.4 | 163.2 | 205.8 | 305.3 | 457.9 | 478.1 | - | |
| The temperature of transformation end, °C | | 131.4 | 172.1 | 217.6 | 318.2 | 473.1 | 481.8 | - | |
| Energy balance, J $\times$ g$^{-1}$ | | $-10.4$ | $-26.8$ | 10.8 | 105.1 | $-58.9$ | $-9.3$ | - | 10.5 |
| The temperature of transformation beginning, °C | | 110.1 | 244.4 | 255.2 | 283.2 | 417.48 | 445.9 | 465.1 | |
| The temperature of transformation peak, °C | Leno | 120.3 | 249.3 | 256.9 | 298.1 | 425.59 | 449.6 | 474.9 | |
| The temperature of transformation end, °C | | 127.9 | 254.6 | 258.3 | 308.0 | 434.95 | 455.3 | 481.7 | |
| Energy balance, J $\times$ g$^{-1}$ | | $-0.92$ | $-16.9$ | $-1.0$ | 164.6 | $-72.7$ | $-31.1$ | $-40.6$ | 1.4 |

DTG and DSC thermal analyses of diapers was carried out by Stępień et al. [45]. Three endothermic transformations up to 300 ºC were observed. For comparison, in this study six transformations up to 500 °C were observed, of which the first two were endoenergetic and the third was exoenergetic up to 230 °C. It was shown in the study of Stępień et al. that diapers required about 1276 J·g$^{-1}$ of energy in the process up to 300 °C. As previously mentioned, the energy determined by Stępień et al. [45] refers to the whole process. Nevertheless, it is worth noting that the energy requirement for diapers is close to those of paper and cardboard.

### 3.3.6. Biodegradable Waste

The TG/DTG curves for poultry meat in Figure 11a and for potato peelings in Figure 11b are presented. The most dynamic weight loss for poultry meat was observed between 72 and 107 min at temperatures of 370–540 °C. For potato peelings, the most dynamic decrease occurred between 65 and 85 min, between 333–430 °C. The weight loss in the case of potato peels (Figure 11a) was related to the thermal decomposition of starch, whereas in the case of chicken it was related to the decomposition of proteins (Figure 11a).

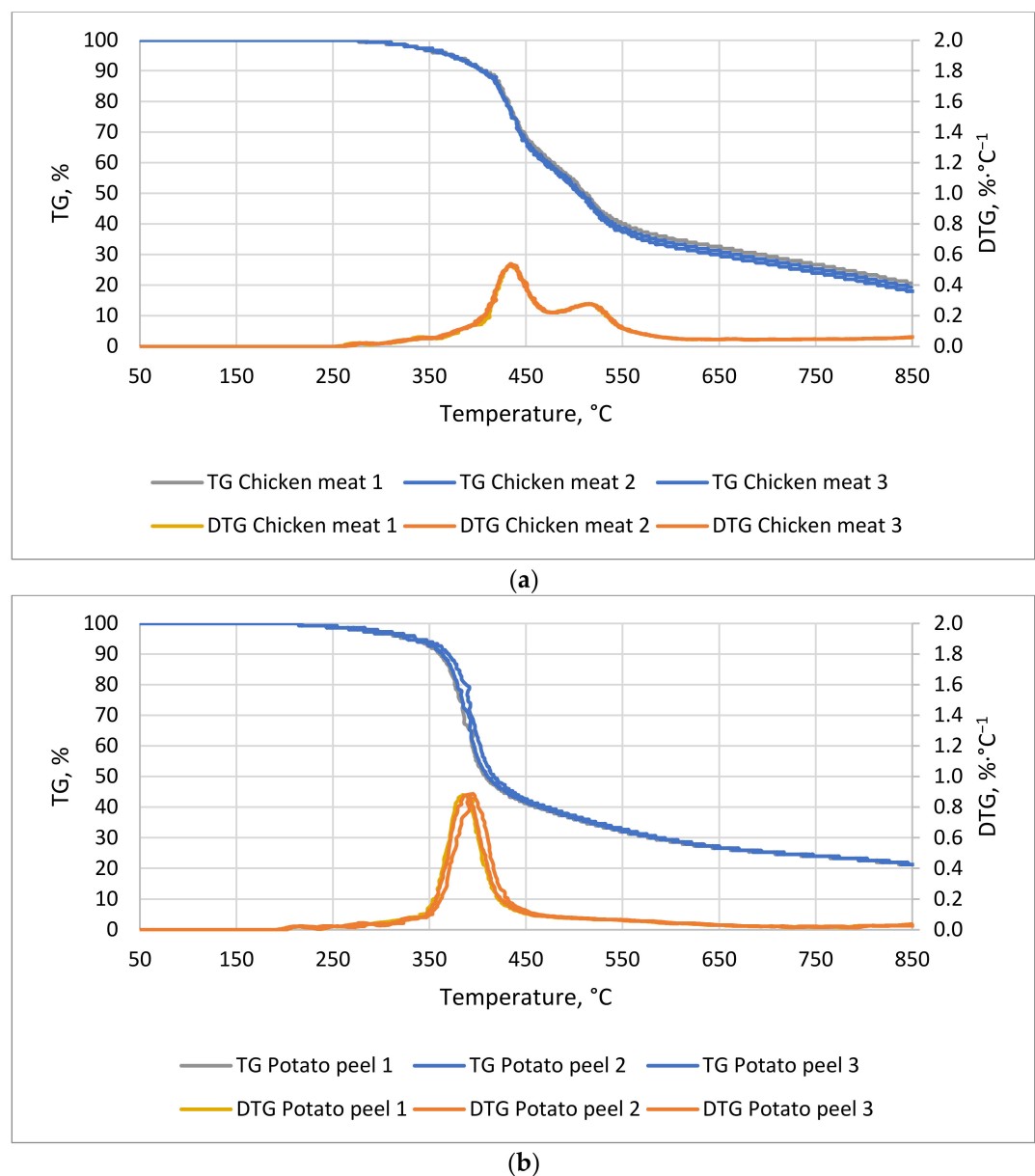

**Figure 11.** Characteristics of the TG/DTG of biodegradable waste: (**a**) chicken meat; (**b**) potato peel.

The first decrease in the weight of the potato peel sample was recorded at about 220 °C. Between 220 and 350 °C, the mass loss was small, i.e., about 6%. In contrast, in the temperature range of 350–450 °C, there was a faster mass decrease (about 45% relative to the mass at 350 °C) [58]. The most dynamic weight loss for potato peelings occurred in a shorter time than for cellulosic materials. This result is related to the degree of polymerization of starch, which is usually lower than the polymerization of cellulose. Further weight loss was about 20% of the initial weight [59].

The main component of chicken meat, other than water, is protein, which constitutes 24%, and fat, which constitutes about 2%. The decrease in weight was mainly related to the decomposition of fats because the proteins in meat decomposed at 59 and 82 °C (myosin, collagen, and sarcoplasmic proteins, actin) [60].

Five endothermic transformations were observed during the DSC analysis of poultry meat (Figure 12a). They occurred in the following temperature ranges: 171–205 °C; 245–278 °C; 281–292 °C; 406–411 °C; and 415–426 °C. Comparing these data with the results of the DTG/TG analysis, there was no mass decrease during the first transformation. This is probably due to the difference between the actual temperature of the reactor and

the temperature of the sample, or because there is a chemical reaction taking place which results in a new product. The first transformation can be related to the transformation of arachidonic acid, whose boiling point is 170 °C. The second transformation may be related to the chemical reaction of erucic acid (boiling point of 265 °C) [58]. As with arachidonic acid, no decrease in weight was observed during DTG/TG analysis. For the subsequent transformations, no links were found with which to associate them. Poultry meat also contains other acids with higher boiling points, i.e., palmitic acid (351 °C), oleic acid (360 °C), and nervonic acid (479 °C), and cholesterol, whose boiling occurs at 360 °C. The transformations occurring at this time are likely related to the decomposition of the charred sample [58].

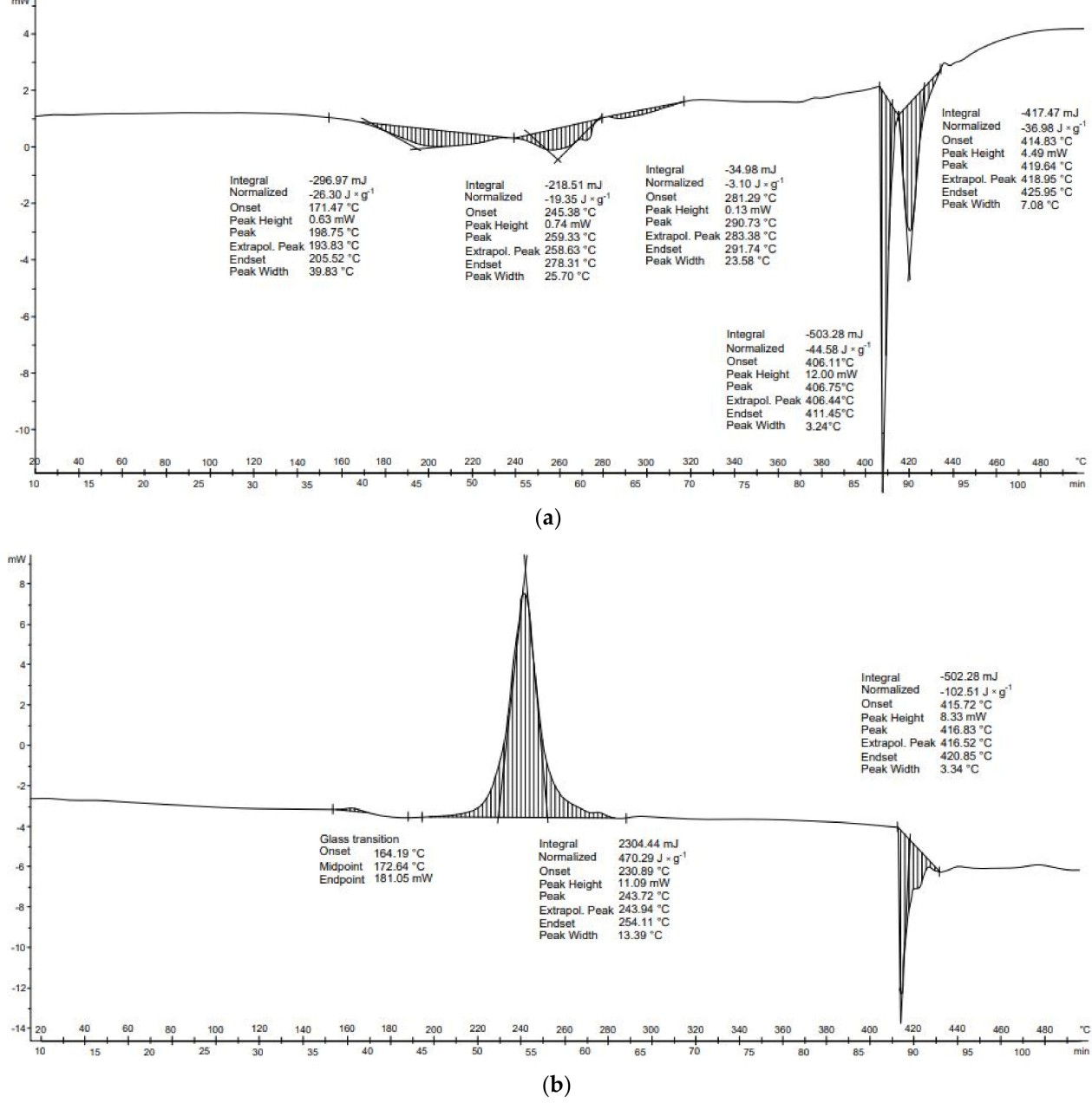

**Figure 12.** Characteristics of the DSC of biodegradable waste: (**a**) chicken meat; (**b**) potato peel.

Potato peelings (Figure 12b) are characterized by three transformations: vitrification (164–181 °C), exothermal transformation (234–254 °C), and endothermal transformation (416–421 °C). The second, exothermal, transformation occurred at temperatures of

231–254 °C. The final transformation, of an endothermal nature, was observed between 416 and 421 °C. Furthermore, no exothermal transformations occurred in any DSC analysis of potato starch. These transformations may be related to the previous drying of the potato sample and thus to the thermal decomposition of the unfolded proteins.

Five transformations occurred in the thermal analysis of chicken meat. The first transformation was endothermic ($-26.3\,\text{J} \times \text{g}^{-1}$), the second transformation was endothermic ($-19.4\,\text{J} \times \text{g}^{-1}$), the third transformation was endothermic ($-3.1\,\text{J} \times \text{g}^{-1}$), the fourth transformation was endothermic ($-44.6\,\text{J} \times \text{g}^{-1}$), and the final transformation was endothermic ($-37.0\,\text{J} \times \text{g}^{-1}$). The energy balance of the thermal transformations of the chicken meat was negative ($-130.3\,\text{J} \times \text{g}^{-1}$) (Table 8). In the case of potato peelings, three thermal transformations occurred. The first transformation occurred but its value equilibrated, the second transformation was exothermic ($470.3\,\text{J} \times \text{g}^{-1}$), and the third transformation was endothermic ($-102.5\,\text{J} \times \text{g}^{-1}$). The overall energy balance of the thermal transformations of the potato peel was positive ($367.8\,\text{J} \times \text{g}^{-1}$).

**Table 8.** Normalized transformation energy for successive transformations of tested biodegradable waste and energy balance of the process.

| DSC Characteristics | Type of Material | Transformation | | | | | | | |
|---|---|---|---|---|---|---|---|---|---|
| | | 1 | 2 | 3 | 4 | 5 | 6 | 7 | |
| The temperature of transformation beginning, °C | Chicken meat | 171.5 | 245.4 | 281.3 | 406.1 | 414.8 | - | - | |
| The temperature of transformation peak, °C | | 198.8 | 259.3 | 290.7 | 406.8 | 419.6 | - | - | |
| The temperature of transformation end, °C | | 205.5 | 278.3 | 291.7 | 411.5 | 426.0 | - | - | |
| Energy balance, $\text{J} \times \text{g}^{-1}$ | | −26.3 | −19.4 | −3.1 | −44.6 | −37.0 | − | − | −130.3 |
| The temperature of transformation beginning, °C | Potato peel | 164.2 | 230.9 | 415.7 | - | - | - | - | |
| The temperature of transformation peak, °C | | 172.6 | 243.7 | 416.8 | - | - | - | - | |
| The temperature of transformation end, °C | | 181.1 | 254.1 | 420.9 | - | - | - | - | |
| Energy balance, $\text{J} \times \text{g}^{-1}$ | | - | 470.3 | −102.5 | − | − | − | − | 367.8 |

The energy balance for transformations of biodegradable wastes showed inconclusive results. During the pyrolysis of chicken meat, it was not possible to balance the energy of transformations because all of them were endothermal. In contrast, to reduce the energy required for pyrolysis of potato peels, temperatures of 220–240 °C should be maintained. According to the studies of Stępień et al. [45], chicken meat pyrolyzed up to 300 °C was characterized by low energy needs of $215\,\text{J} \cdot \text{g}^{-1}$. In this study, the energy balance of transformations for processes up to 500 °C was $-130.3\,\text{J} \times \text{g}^{-1}$. These results show a different trend than that observed by Stępień et al. [45] because, as there is no exothermal transformation, the energy for the process should be significantly higher. This phenomenon may occur because chicken meat has a significantly lower specific heat than those of the previously mentioned wastes. As a result, the total energy required to heat chicken meat is significantly lower than that required for cardboard and paper wastes, despite the lack of exothermal reactions in chicken meat. Nevertheless, this phenomenon needs to be confirmed by additional experiments.

## 4. Conclusions

Twelve main organic components of municipal solid waste were subjected to proximate analysis and TG/DTG/DSC thermal analysis.

The proximate analysis showed that the greatest moisture waste was found in the biodegradable group (chicken residues and potato peel), with moisture content over 50%. The driest materials were found in the plastic group, namely PET and PU. PET also had the highest calorific value of $42\,\text{MJ} \times \text{kg}^{-1}$.

Among all of the tested wastes, plastic materials showed the greatest differences in weight loss. The PET material had the highest weight loss, of around 91%, whereas the PU had the lowest weight loss, of around 60%. In the case of the remainder of the tested groups, the differences between material weight losses were significantly less visible, and for the hygienic and biodegradable waste groups, the differences were small.

Kinetics analysis was performed using the Coats–Redfern method at a heating rate of 5 °C × min$^{-1}$. For the tested materials, the activation energy varied from 25.2 to 134.5 kJ × (mol × K)$^{-1}$ for leather and cotton, respectively. The mane vale was found to be 74.3 kJ × (mol × K)$^{-1}$. The reaction order differed significantly for each material, from 1.41 to 3.52 for diapers and receipts, respectively.

Based on DSC analysis, the balance of thermal transformations during pyrolysis was calculated, thus increasing the understanding of the transformations occurring during the thermal treatment of waste. For all tested materials, the highest positive energy balance of transformations occurred for potato peels (367.8 J × g$^{-1}$), whereas the lowest negative balance was determined for PET plastic (−220.1 J × g$^{-1}$). The main results of the energy balance of transformations occurring during pyrolysis up to 500 °C are shown in Figure 13. Only three of the tested materials showed a negative energy balance for transformations, namely leather, PET, and chicken meat. Although these materials have negative energy of transformations, energy is required for the pyrolysis of these materials due to their specific heat [45]. Nevertheless, this study showed that energy needs for pyrolysis can be minimalized by only heating materials to a temperature at which exothermal reactions occur. For most of the tested materials, this was between 220 and 340 °C. Therefore, this temperature range is recommended for the pyrolysis of municipal solid waste to minimize pyrolysis energy consumption.

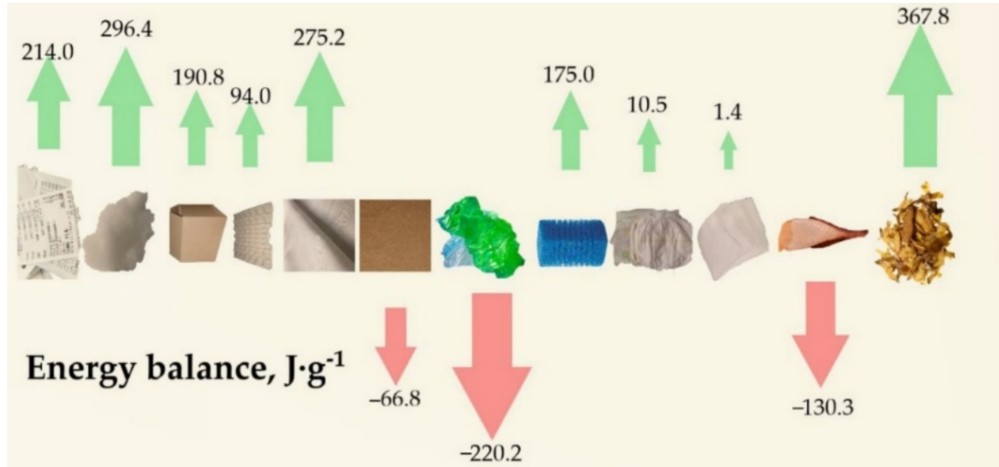

**Figure 13.** Energy balance of transformations occurring during pyrolysis for different wastes.

Information regarding the point at which endoenergetic or exoenergetic transformation takes place in waste leads to the determination of optimal conditions for transforming waste into alternative fuels. For some materials in which high energy-consuming transformations occur at high temperatures, pyrolysis may not be justified. Nevertheless, this study needs to be extended to the fuel analysis of products of low-temperature pyrolysis, because this would allow determination of the quality of the fuel related to the energy needed for its production.

This study provided information regarding the physical properties of main organic wastes and their thermal behavior during low-temperature pyrolysis. The results showed that each component has different properties and thermal degradation behaviors. The determination of these properties and the acquisition of related knowledge are essential for the pyrolysis of municipal solid fractions.

**Author Contributions:** Conceptualization, A.B., E.S. and K.Ś.; methodology, A.B. and I.K.; validation, A.B., E.S. and K.Ś.; formal analysis, A.B., E.S. and K.Ś.; investigation, I.K.; resources, A.B. and I.K.; data curation, E.S. and K.Ś.; writing—original draft preparation, E.S. and K.Ś.; writing—review and editing, A.B., E.S., M.H. and K.Ś.; visualization, M.H., E.S. and K.Ś.; supervision, A.B. All authors have read and agreed to the published version of the manuscript.

**Funding:** This research received no external funding. The publication is financed under the Leading Research Groups support project from the subsidy increased for the period 2020–2025 in the amount of 2% of the subsidy referred to Art. 387 (3) of the Law of 20 July 2018 on Higher Education and Science, obtained in 2019.

**Institutional Review Board Statement:** Not applicable.

**Informed Consent Statement:** Not applicable.

**Data Availability Statement:** All data derived during the experiments are given in the paper.

**Acknowledgments:** The presented article results were obtained as part of the activity of the leading research team—Waste and Biomass Valorization Group (WBVG).

**Conflicts of Interest:** The authors declare no conflict of interest.

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
