# Peer review of "Municipal Solid Waste Thermal Analysis—Pyrolysis Kinetics and Decomposition Reactions"

_energies, doi:10.3390/en14154510_

Round 1

Reviewer 1 Report

The article 'Municipal Solid Waste Thermal Analysis – Pyrolysis Kinetics 
and Decomposition Reaction' is interesting, however, the authors should improve the introduction, and put more emphasis on explaining the results. The introduction does not provide any background info, but it is rather a recap of the definitions, which are well established/known in the literature.  

Reviewer 2 Report

The paper is related to waste analysis in Poland. The writing is well and the paper has a proper structure. Some issues should be considered before publication.

Types of wastes should be explained in the abstract. 

the introduction is too general can be more specific. I can see much information regarding the used waste while the writing regarding parameters and processes is too much. It is better to make a balance in the content. The citation in this section can be up to 25. Now only 15 references are acceptable which is too low.

Lines 47-58: the information regarding TGA is too much and should be summarized. 

What are the citations in line 80? [16-19]? I think is it a mistake by the authors. Here you should explain your novelty in clearer form.

The novelty of this research is not understandable to the readers now. Why this paper should be published and how this data can help other future studies and Poland?

What is the difference between this study rather than published studies in this field? Are there any previous studies regarding waste analysis in Poland? I suggest citing them and compare your results with those studies.

The discussion section should be enriched by more cited studies that have been published recently (2018-2021). 

Lines 554-559: are not necessary and can be deleted.

Reviewer 3 Report

The main drawback of the paper are the following:

The Literature review or Background section is missing and I suggest splitting the Introduction section in two sections – one Introduction and the other Literature Review.

The research approach should include the methods employed as well as a real dialog between literature and methods.

The discussion does not well integrate the literature with the results. This needs to be addressed in order to provide a more coherent argument in favor of the results.

I suggest providing more depth to your conclusion to improve its efficacy and relation to the rest of the paper.

The Table 1. Research materials from six basic groups of waste is rather simplistic and can be explained as text in the article.

It is necessary to pay attention to the quality of the figures that is very low.

Good luck!

Reviewer 4 Report

Тhe paper presented for the review is dedicated to the estimation of municipal solid waste by thermal analysis, pyrolysis kinetics and decomposition reactions. The theme of the work is suitable for the Energies Journal.

The title of the paper provides enough information and reflects the content of the work.

The work structure is adequate; in the introduction, the authors gave enough background information on the current state of scientific research in the subject area of the article. However, there is a lack of details about the influence of pyrolysis on the environment that needs to be covered.

The methods and materials should be explained in more detail. On page 4, the equations should be numbered and segregated from the text.

There is a problem in the results and discussion section with the visibility and quality of the figures. All of them are unreadable and blurry. The authors should provide better quality figures and enlarge them so all the numbers and the text can be seen. There is a typing mistake; figure 9 appears twice, figure 8 is missing (on page 15 and page 16). On page 19, line 492, the temperatures are given with a comma instead of a dot and without °C symbol.

The discussion does not well integrate the literature with the results.

The given tables are descriptive and elaborative in the function of the presented results. Still, beneath table 2, a legend with the explained abbreviations should be provided. The names of research materials should be consistent thought the text; check table 2. Table 1.is slightly simplified and can be described as text in the article.

The conclusions section summarises the research and gives the essence of the presented results. The study highlighted the importance of knowing the type of processed waste material thus the different conditions for achieving equal conversion. The paper opens the possibility for future research to focus on mixed municipal waste pyrolysis optimization.

The chosen references correspond to the article topic.

Round 2

Reviewer 3 Report

Good luck!

Reviewer 4 Report

After examining the corrected manuscript, the problems have been altered and explained satisfactorily. Accordingly, I would like to recommend it for publishing.